# Rewriting nuclear epigenetic scripts in mitochondrial diseases as a strategy for heteroplasmy control

María J Pérez[1,2,7], Rocío B Colombo [1,7], Sebastián M Real [1,3], María T Branham[1,4], Sergio R Laurito [1,5], Carlos T Moraes [6] & Lía Mayorga [1✉]

## Abstract

Mitochondrial diseases, caused by mutations in nuclear or mitochondrial DNA (mtDNA), have limited treatment options. For mtDNA mutations, reducing the mutant-to-wild-type mtDNA ratio (heteroplasmy shift) is a promising strategy, though it currently faces challenges. Previous research showed that severe mitochondrial dysfunction triggers an adaptive nuclear epigenetic response, through changes in DNA methylation, absent or less important for subtle mitochondrial impairment. Therefore, we hypothesized that targeting nuclear DNA methylation could impair cells with high-mutant mtDNA load while sparing those with lower levels, reducing overall heteroplasmy. Using cybrid models harboring two disease-causing mtDNA mutations—m.13513 G > A and m.8344 A > G—at varying heteroplasmies, we discovered that both the mutation type and load distinctly shape the nuclear DNA methylome. We found this methylation pattern critical for the survival of high-heteroplasmy cells but not for low-heteroplasmy ones. Treatment with FDA-approved DNA methylation inhibitors selectively impacted high-heteroplasmy cybrids and reduced heteroplasmy. These findings were validated in cultured cells and xenografts. Our findings highlight nuclear DNA methylation as a key regulator of heteroplasmic cell survival and a potential therapeutic target for mitochondrial diseases.

**Keywords** Mitochondrial Diseases; Heteroplasmy; Mitochondrial DNA; DNA Methylation; Epigenetics
**Subject Categories** Chromatin, Transcription & Genomics; Genetics, Gene Therapy & Genetic Disease; Organelles

## Introduction

Mitochondrial diseases impact approximately 1 in every 5000 individuals, making them among the most prevalent inherited metabolic disorders (Gorman et al, 2015; Thorburn, 2004; Schaefer et al, 2008). To date, most of these disorders have inefficient or no treatment available. These diseases can arise from mutations in either nuclear or mitochondrial DNA (mtDNA). While traditional gene therapy approaches are becoming clinically available for nuclear genetic disorders (Sayed et al, 2022; Elangkovan and Dickson, 2021; Rossoll and Singh, 2022; Shahryari et al, 2019), manipulating the mitochondrial genome has proven more difficult due to its unique properties. mtDNA is small, circular, and exists in multiple copies within the mitochondrial matrix. It encodes for 13 essential respiratory chain subunits, 2 ribosomal RNAs and 22 transfer RNAs, necessary for mitochondrial translation. There are no known RNA import systems to the mitochondria (Gammage et al, 2017; Schmiderer et al, 2022), so base editing the mitochondrial genome has been challenging. However, new protein-only technologies have emerged for mtDNA editing (Moraes, 2024; Mok et al, 2020). Because mtDNA mutations coexist with wild-type mtDNA (mtDNA heteroplasmy), the percentage of mutated molecules defines the emergence of symptoms only when higher than a specific threshold (DiMauro and Moraes, 1993). Therefore, reducing the mutant-to-wild-type mtDNA ratios or "heteroplasmy shift" is a valuable strategy for treatment. Although different techniques targeting the mtDNA for heteroplasmy shift have been successful in laboratory settings (mitochondrially targeted restriction endonucleases (Srivastava and Moraes, 2001), zinc finger nucleases (Gammage et al, 2014), mitoTALENs (Bacman et al, 2013), mitoArcus (Shoop et al, 2023)), none have yet reached clinical practice as gene therapy faces several challenges (Falabella et al, 2022).

In this study, we propose a novel approach to lower heteroplasmy taking advantage of the natural communication between mitochondria and the nucleus, particularly the nuclear epigenome. Mitochondrial metabolites, such as succinate, fumarate, NAD$^+$, NADH and reactive oxygen species, modify the function of epigenetic enzymes (Jumonji C, TETs, DNA methyltransferases) (Zhou et al, 2016; Peng et al, 2011; Kreuz and Fischle, 2016). Consequently, different degrees of mitochondrial stress modify nuclear transcriptomics (Picard et al, 2014), histone marks (Kopinski et al, 2019), and DNA methylation (Mayorga et al, 2019) in a specific manner. The latter has been proposed by us as an adaptive strategy to intense, chronic mitochondrial stress (Mayorga et al, 2019). We previously showed

[1]Instituto de Histología y Embriología de Mendoza (IHEM), Universidad Nacional de Cuyo, CONICET, Mendoza, Argentina. [2]Facultad de Ciencias de la Nutrición. Universidad Juan Agustín Maza, Mendoza, Argentina. [3]Instituto de Fisiología, Facultad de Ciencias Médicas. Universidad Nacional de Cuyo, Mendoza, Argentina. [4]Facultad de Ciencias Médicas. Universidad de Mendoza, Mendoza, Argentina. [5]Facultad de Ciencias Exactas y Naturales. Universidad Nacional de Cuyo, Mendoza, Argentina. [6]Department of Neurology, University of Miami Miller School of Medicine, Miami, FL, USA. [7]These authors contributed equally: María J Pérez, Rocío B Colombo.
✉E-mail: liamayorga@fcm.uncu.edu.ar

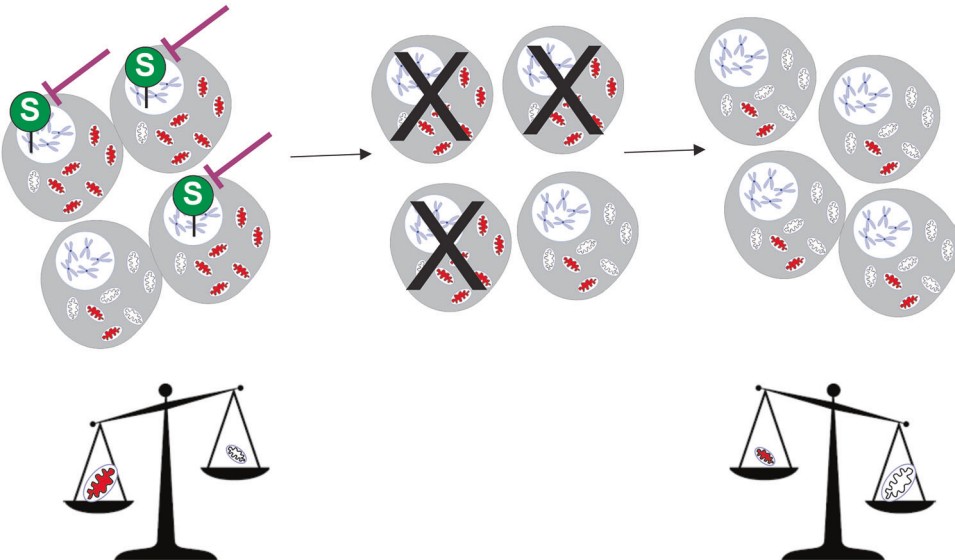

**Figure 1. Working hypothesis.**

Mitochondrial dysfunction impacts the nuclear epigenome. Particularly, intense mitochondrial dysfunction determines a "survival" nDNA methylation patterning (Mayorga et al, 2019). Considering that high heteroplasmy implies intense mitochondrial impediment, these cells would have the "S" nDNA methylation design, whereas low-heteroplasmy cells would not. By erasing this "S" pattern, the survival of the high-heteroplasmy cells would be impaired, favoring the low-heteroplasmy cells and therefore lowering heteroplasmy. Red mitochondria = mitochondria with mainly mutated mtDNA, white mitochondria = mitochondria with mainly wild-type mtDNA molecules. A green circle with a white "S" represents a "survival" nDNA methylation pattern.

that severe mitochondrial dysfunction triggered an epigenetic pro-survival response through modifications of the nuclear DNA (nDNA) methylation pattern, a phenomenon that did not occur when mitochondrial dysfunction was subtle (Mayorga et al, 2019). Thus, considering that high-heteroplasmy is associated with severe mitochondrial respiration impairment, these cells may depend on an adaptive strategy which includes changes in nDNA methylation, a mechanism which would not be triggered in low-heteroplasmy cells. We hypothesized that altering the nDNA methylome would impair high-heteroplasmy cells, promote the dominance of low-heteroplasmy cells, and thereby lead to an overall reduction in heteroplasmy (Fig. 1). Notably, DNA methylation modulators like 5-Azacytidine and Decitabine are already FDA-approved for other conditions (Gawlitza et al, 2019; Christman, 2002), raising the possibility of repurposing these drugs for mtDNA disorders.

## Results

As a model of mitochondrial diseases, we used transmitochondrial cybrids (Bacman et al, 2020; Hashimoto et al, 2015) harboring wild-type mtDNA and two disease-causing mutations: m.13513 G > A (MT-ND5) (Shanske et al, 2008; Wang et al, 2010, 2023) and m.8344 A > G (MT-tRNA-Lys) (Silvestri et al, 1993; Shoffner et al, 1990). The ND5 mutation is associated with Leigh syndrome (Sudo et al, 2004), MELAS (Shanske et al, 2001) and LHON (Andreeva et al, 2022) phenotypes. The m.8344 A > G mutation is most often associated with the MERRF (Zeviani et al, 1991; Silvestri et al, 1993) phenotype. Cell clones with high or low mtDNA mutant loads, based on previously defined phenotypic thresholds or severities (Moslemi et al, 1998; Finsterer, 2023; Andreeva et al, 2022), (see "Methods" for

specific heteroplasmies) were selected for further experiments. For brevity, we refer to cybrids harboring the m.13513 G > A mtDNA mutation as "13" followed by "L" for low-mutant load and "H" for high-mutant load. Likewise, the m.8344 A > G cybrids are referred to as 8L and 8H. The distinct mitochondrial functioning between high and low-heteroplasmy cybrids was validated through differences in mitochondrial membrane potential (Fig. EV1).

## The nuclear DNA methylome is shaped by both the degree of mtDNA heteroplasmy and the nature of the mtDNA mutation

### DNA methylation of tumor suppressor genes

Since our hypothesis states that DNA methylation could be important for the survival of high-heteroplasmy cells, we analyzed the methylation status of CpG sites located in promoter regions from tumor suppressor genes that are commonly hypermethylated in cancer (paradigm of cellular survival) (Buyru et al, 2009; Maleva Kostovska et al, 2018; Furlan et al, 2013). We investigated 52 CpG sites using MS-MLPA (Methyl-specific multiple ligation-dependent probe amplification) (Nygren et al, 2005) in promoter regions of 35 tumor suppressor genes in wild-type and mutated cybrids.

Unsupervised hierarchical cluster using Manhattan's distance and average linkage clustering method of the mean methylation percentages of each group was performed to demonstrate differences or similarities between the groups (Fig. 2A). When comparing paired CpG-by-CpG mean methylation levels, there was a greater methylation of these regions in the 13H cybrids compared to the 13L and wild-type ones (Fig. 2B). This is in accordance with our previous work, in which we had observed an increased methylation of these sites (plus others) in samples coming from

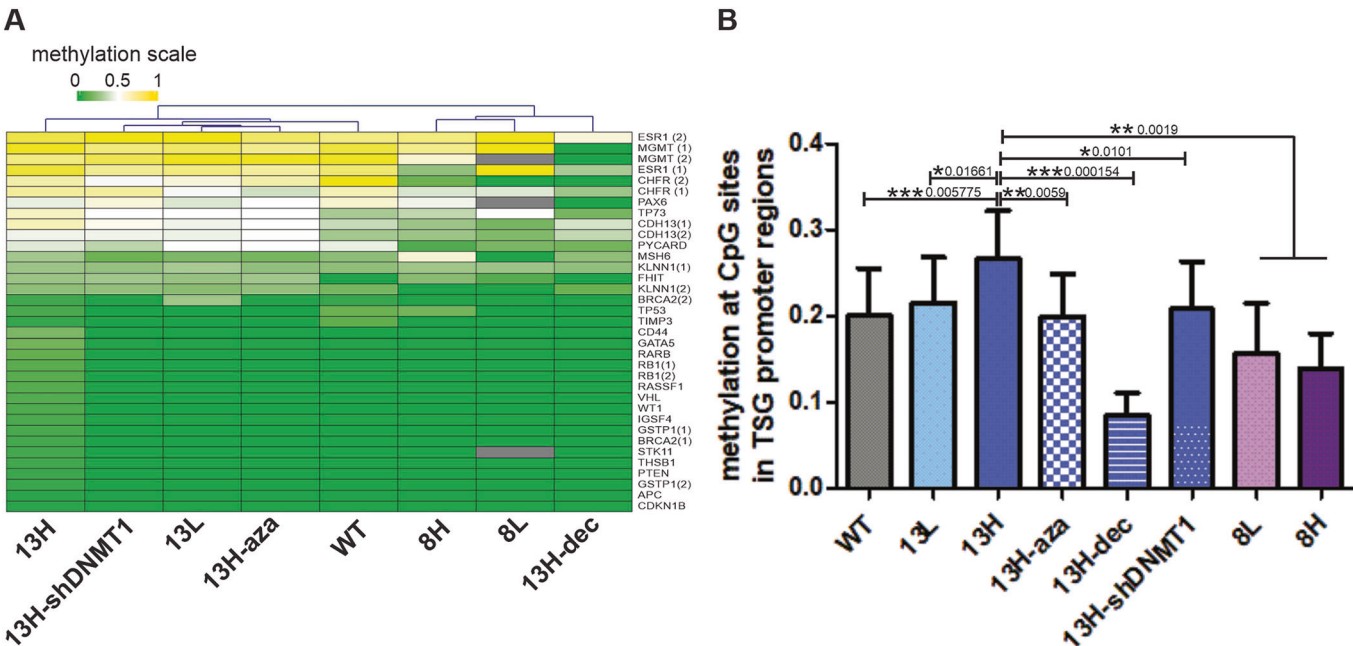

**Figure 2. DNA methylation of CpG sites in promoter regions of tumor suppressor genes (TSG) differs between groups.**

Cybrid cells with different mtDNA mutations and heteroplasmies were analyzed using MS-MLPA. (A) Heatmap and hierarchical cluster of 35 CpG sites in cybrid samples using Manhattan's distance and average linkage clustering method of the mean methylation percentages of each group. Color scale: green, methylation=0%, yellow, methylation=100% (1), white, methylation = 50% (0.5), gray = not analyzed: N/A. Column cluster dendrogram is shown, row clustering was performed but dendogram is not shown. Mean methylation of each CpG site from the different sample groups was used for analysis. CpG sites that had 0% methylation in all groups were excluded from analysis, 35 out of 52 remained for analysis. (B) Comparison of the mean methylation level of TSG promoter regions. Paired CpG-by-CpG comparison from 32 CpG sites—3 sites with missing values were eliminated from analysis—(mean methylation of each CpG site from the different sample groups was used for the Friedman test + Dunn's Multiple Comparison Test). ***$P < 0.001$, **$P<0.01$, *$P<0.05$. For this figure in general: WT wild-type cybrids, 13L low-heteroplasmy m.13513 G > A cybrids, 13H high-heteroplasmy m.13513 G > A cybrids, 8L low-heteroplasmy m.8344 A > G cybrids, 8H high-heteroplasmy m.8344 A > G cybrids, aza 5-azacytidine, dec decitabine, shDNMT1 IPTG-Inducible shRNA DNMT1 knockdown. $N = 4$ samples for condition for 13H, 2 samples for WT, 13L, 13H-shDNMT1, 8H, 8L, and 1 for 13H-aza. Samples from each group came from different passages. Bars represent mean +/− SEM. Source data are available online for this figure.

mitochondrial disease patients (Mayorga et al, 2019). The "8" cybrids did not differ between the high and low-heteroplasmy group regarding these regions but did exhibit a generalized lower methylation level compared to the "13" cybrids (Fig. 2B). Therefore, at least for the "13" mutants, in line with our hypothesis, as heteroplasmy levels increase, CpGs related to tumor suppressor genes show greater methylation, which in theory, would enhance the survival capacity of the high-heteroplasmy cells.

In addition, we tested different strategies to reduce DNA methylation levels globally. For this, we treated 13H cells with DNA methylation inhibitors: 5-azacytidine and decitabine. Furthermore, we knocked down the expression of the DNA methyltransferase 1 (DNMT1, knockdown control in Fig. EV2), necessary for maintaining the DNA methylation pattern (Davletgildeeva and Kuznetsov, 2024). All three approaches decreased the methylation status of these CpG sites with decitabine being the most effective to do so (Fig. 2B).

### Genome-wide methylation

Given the changes observed in a limited number of CpG sites, we expanded our analysis to investigate global DNA methylation. Using Illumina's Infinium EPIC 850k array (Pidsley et al, 2016), which covers over 850,000 CpG sites across the genome, we examined genome-wide methylation patterns of wild-type, 13H, 13L, 8H, and 8L cybrids.

Unsupervised hierarchical clustering of these methylation profiles (using Manhattan's distance and average linkage) revealed that the samples clustered according to mutation type and load. This suggests that both the form and intensity of mitochondrial dysfunction are key determinants of the DNA methylation profile (Fig. 3A).

Mean beta-values (which represent the methylation level at each CpG site) (Pidsley et al, 2016) across the genome exhibited increased levels of DNA methylation in high-heteroplasmy cells (Fig. 3B, left panel), in accordance with our previous work with patients' muscle samples (Mayorga et al, 2019), and with the methylation level of tumor suppressor genes described above. In addition, the "13" mutant cybrids globally showed higher methylation than the "8" lines, and higher than the wild-type cells (Fig. 3B, right panel), again in line with what we had observed in tumor suppressor genes. Importantly, the differentially methylated CpG sites followed the same global methylation trend. When comparing mutant versus wild-type cybrids (Fig. 3C, left panel), most differentially methylated sites were hypermethylated, and the same pattern was observed when comparing high- versus low-heteroplasmy cybrids (Fig. 3C, right panel). Sample 8L_1, with a borderline heteroplasmy level (~50%), exhibited a closer association with the high-mutant samples. Beta-values for each site and sample are available in Dataset EV1.

While the implications of nuclear DNA methylation are well-established (Deaton and Bird, 2011; Adrian, 2017; Razin and Riggs,

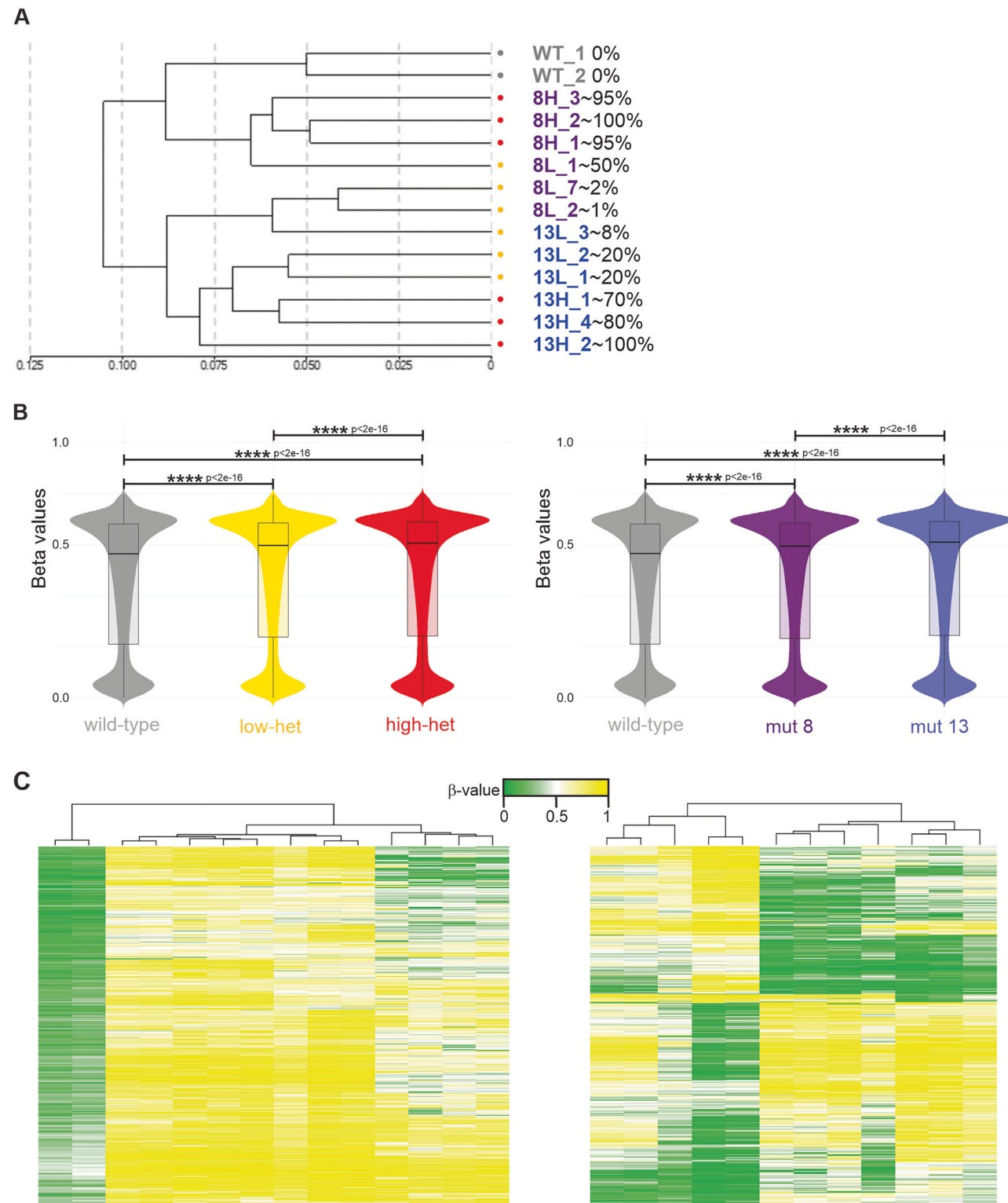

**Figure 3.   Genome-wide methylation pattern differs between groups. Analysis of over 850,000 CpG sites using Illumina's Infinium EPIC 850k.**

(A) Full methylation pattern sample dendrogram obtained through Illumina's GenomeStudio Software®. Unsupervised hierarchical cluster using Manhattan's distance and average linkage clustering method. Next to each sample name, the approximate heteroplasmy percentage is shown. The samples cluster according to mutation type (or wild-type) and heteroplasmy. (B) Violin plots and Box plot graphs generated from average beta-values of each subgroup. Left panel comparison according to heteroplasmy subgroups ($N = 2$ wild-type, 6 low-heteroplasmy, 6 high-heteroplasmy samples). Right panel comparison between mutation types ($N = 2$ wild-type, 6 m.8344 A > G and 6 m.13513 G > A samples). Wilcoxon matched-pairs signed rank test for paired comparisons of mean values at each CpG site. Horizontal lines denote the median, box limits indicate 25th and 75th percentiles and whiskers extend from the lowest/largest value. (C) Heatmap showing Unsupervised hierarchical clustering (Manhattan's distance + average linkage) of the top significantly ($P < 0.05$) differentially methylated regions (DMRs). Δ mut-wt >0.4 or <−0.4 (left panel) and Δ high-low >0.27 or <−0.27. Column cluster dendrogram is shown, row clustering was performed but dendogram is hidden. Throughout this figure, samples are highlighted with different colors according to mutation type (gray = wild-type, blue= m.13513 G > A, purple= m.8344 A > G) and heteroplasmy (gray= 0%, gold= low-heteroplasmy, red= high-heteroplasmy). Source data are available online for this figure.

1980; Esteller et al, 2024), the extent, nature, and significance of mtDNA methylation remain controversial (Tong et al, 2017; Stoccoro and Coppedè, 2021). Given our focus on mtDNA mutations and mitochondrial dysfunction, we did not forget to look at mtDNA methylation. Despite its underrepresentation in the EPIC array (with only 7 probes among over 850,000), none of these mtDNA CpG sites showed differential methylation in any of our comparisons (mutant vs. wild-type, high vs. low heteroplasmy, mutant 13 vs. mutant 8; Dataset EV1). This suggests that the significant differential methylation profile we observed is driven by nuclear DNA changes.

## The nDNA methylation profile contributes to explaining the distinct fate behaviors of high and low-heteroplasmy cells

Given the differences observed in the global nDNA methylation analyses above, we aimed to examine the behavior of these cybrids in the presence or absence of DNA methylation modulators.

Intriguingly, high-mutant cybrids showed a higher proliferation rate than their low-heteroplasmy counterparts, a phenomenon which was abolished when DNA methylation was perturbed (Figs. 4A and EV3). Furthermore, high-heteroplasmy cells were more sensitive to apoptosis upon treatment with 5-azacytidine or decitabine (Fig. 4B). To reinforce the importance of DNA methylation for this observation, we also checked the apoptosis level of the m.13513 G > A DNMT1 knockdowns and saw similar effects. The 13L mutants did not increase their apoptotic rate in the presence of demethylating agents nor DNMT1 knockdown. The 8H mutants showed an overall increased apoptosis level compared to the 13H mutants (Fig. 4B).

These data suggest that high-heteroplasmy cells depend on their nDNA methylation marks to proliferate and avoid apoptosis.

To understand the DNA methylation marks that could be contributing to this phenotype, we went back to our EPIC 850k beta-values and compared the high versus low heteroplasmy´s epigenetic signature (for both "13" and "8" mutants). We found that CpG sites with differential methylation were associated to genes with an enriched participation in "survival" pathways (Fig. 5A; Dataset EV1). Genes in these pathways were mostly hypermethylated as had happened across the genome. As a representative of this functional group, ESR1 has been described as a tumor suppressor gene in various cancers, including osteosarcomas (Carvalho et al, 2008; Eads et al, 2001). In these cancers, ESR1's promoter is hypermethylated and therefore underexpressed (Reinert et al, 2022; Osuna et al, 2019). In addition,

its expression is enhanced by decitabine treatment, leading to a reduction in both proliferation and metastasis (Reinert et al, 2022). Since the cybrids we worked with have an osteosarcoma nuclear background (143B), and ESR1 was in the list of hypermethylated genes in high-heteroplasmy cells (Fig. 5A) we looked in detail at the differential methylation of the ESR1 locus (Fig. 5B) and compared the ESR1 expression, namely the estrogen receptor 1 protein between the 13H and 13L lines (Fig. 5C). We found hypermethylation of the whole ESR1 locus (Fig. 5B), a trend to lower expression in high-heteroplasmy cells and a significant increase after treatment with decitabine (Fig. 5C). Such treatment appeared to have no or the opposite effect in the 13L line. Thus, the variation in the abundance of ESR1 (among other differentially methylated genes) could be contributing to the distinctive growth and survival behavior of heteroplasmic cells.

## DNA methylation modulators influence cybrid cells' mitochondrial function

Because these cybrid cells have an altered metabolic state to begin with, we studied whether it was influenced further by DNA methylation modulators. We obtained interesting results regarding mitochondrial function. Firstly, we analyzed the oxygen consumption rate of cells treated with decitabine. We tested 13H, wild-type cybrids and mutated mouse embryonic fibroblasts (m.5024 C > T) with intermediate heteroplasmy (~50%). Surprisingly, decitabine increased their oxygen consumption rate (OCR) and ATP production in all tested cell types (Figs. 6A,B and EV4A), in accordance with previous findings in tumor cells without mtDNA mutations (Ni et al, 2022; Zhu et al, 2020). However, a trend to increased proton leakage (Fig. 6B) and a decrease in mitochondrial membrane potential was also observed (Fig. EV4B), as reported in other cell models (Fandy et al, 2014). The expression of oxidative phosphorylation complex proteins remained unaffected across the conditions (Fig. EV4C).

Considering that these cybrids have impaired mitochondrial function, we reasoned that oxygen may be mishandled and more so by the most dysfunctional cells. In addition, decitabine has previously been shown to increase reactive oxygen species (ROS) in cells with normal mtDNA (Fandy et al, 2014). We investigated the levels of reactive oxygen species (ROS) of the different cybrids with and without treatments. Indeed, decitabine increased the level of ROS for the 13H and 13L cybrids and showed a tendency to do the same in the 8H and 8L cells (Fig. 6C). Knockdown of DNMT1 was not enough to mimic decitabine´s action. Hence, even though oxygen consumption and ATP production increased with DNA methylation inhibition, decitabine treatment led to enhanced ROS

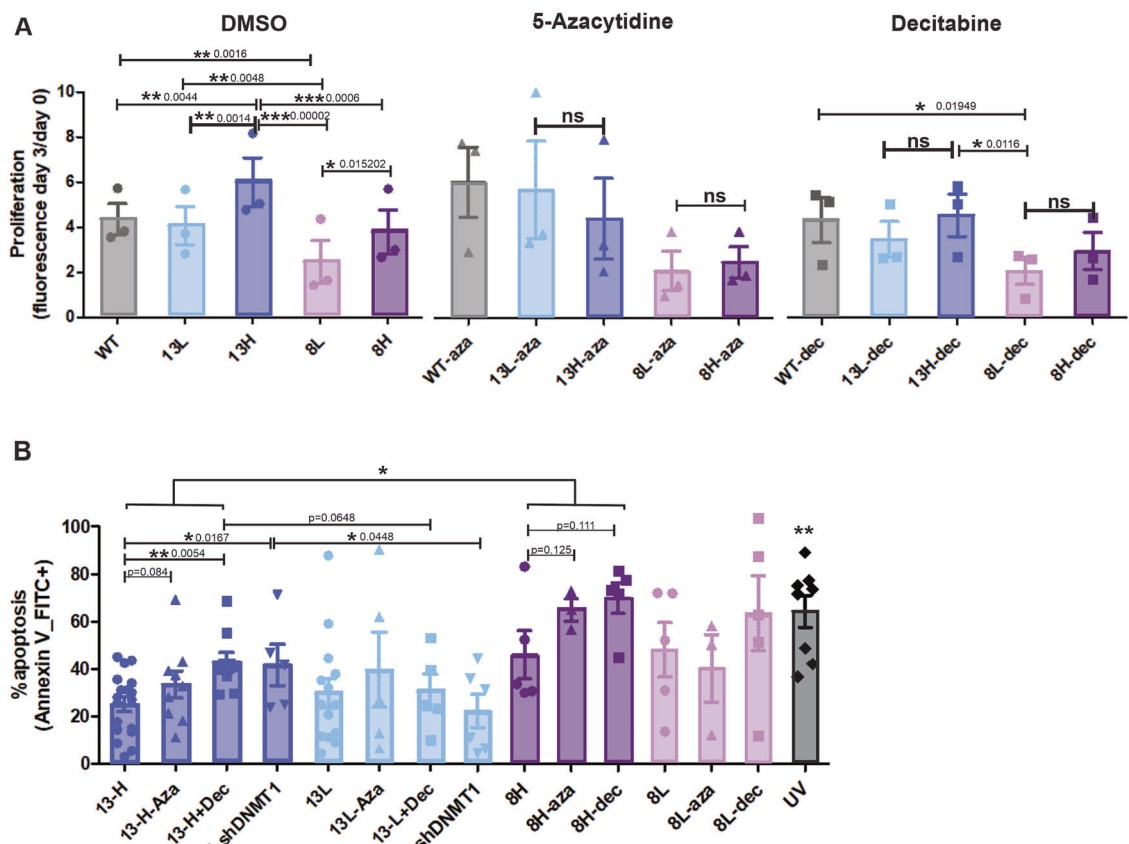

**Figure 4. The nDNA methylation profile determines differential proliferation and apoptosis of cybrids.**

(A) Differential proliferation of cells treated with vehicle (DMSO, left panel), 5-azacytidine (aza) 1 μM—3 days (middle panel), decitabine (dec) 1 μM—3 days (right panel). Other drug concentrations with similar effects are shown in Fig. EV3. **$P < 0.01$, ***$P < 0.001$. Two-way ANOVA with fixed categorical variables, followed by Tukey's Honest Significant Difference test for post hoc comparisons. Residual normality was assessed using the Shapiro–Wilk test, implemented with the tidyr and nortest packages in R. All conditions passed the normality test. $N = 3$ independent biological experiments, with technical duplicates. (B) Apoptosis assay with Annexin V-FITC, Flow cytometry. Using a threshold based on autofluorescence in the Annexin V-FITC histogram, cells were designated as apoptosis ($+$) or ($-$) for quantification. *$P < 0.05$, **$P < 0.01$, one-tailed, unpaired Student´s $t$ tests. $N \geq 3$ independent experiments. Positive control: UV-induced apoptosis. **$P < 0.01$ positive control versus sample with the highest baseline apoptosis level (8L). Bars represent mean $+/-$ SEM. Source data are available online for this figure.

and reduced mitochondrial membrane potential, more pronounced in the high-heteroplasmy cells.

In conclusion, decitabine affects mitochondrial function parameters, with some changes indicating healthier mitochondria and others pointing in the opposite direction. This metabolic dysregulation may be especially detrimental to cells with high heteroplasmy, which—combined with the loss of DNA methylation survival marks—could contribute to their increased susceptibility to apoptosis following decitabine treatment.

## DNA methylation inhibitors lower mtDNA heteroplasmy in cultured cells

Given the notable nDNA methylation difference between cells with varying heteroplasmies and the divergent behavior upon treatment with DNA methylation modulators, we ventured to see if heteroplasmy could be modified when cells were treated with these methylation erasers. Given that the most pronounced differences were in the "13" cybrids, we chose that line for the following

experiments. Mutant mtDNA ratios diminished ~15% consistently in 13H cybrids treated with decitabine (Fig. 7A). Mitophagy was not enhanced by the drugs (Fig. EV5A), nor did mtDNA content vary significantly (Fig. EV5B).

In tissues from patients with mitochondrial diseases high and low-heteroplasmy cells can coexist (Kärppä et al, 2005; Moslemi et al, 1998). So, taking into account our previous results, in which high-heteroplasmy cybrids selectively died and lost their proliferative advantage upon treatment with demethylating agents, we hypothesized that there could be a *selection* toward the low-heteroplasmy cells when treated with these drugs. Maybe the 13H cybrids were not as pure of a clone and had a proportion of 13Ls which made their heteroplasmy sensitive to DNA methylation modulation.

To test this, we deliberately mixed 13H and 13L cybrids before treating them with DNA methylation inhibitors. In addition, we labeled the 13L cells with a green dye in order to trace the percentage of them upon DNA methylation inhibition. When the mixture of cells was treated with the drugs, the percentage of green

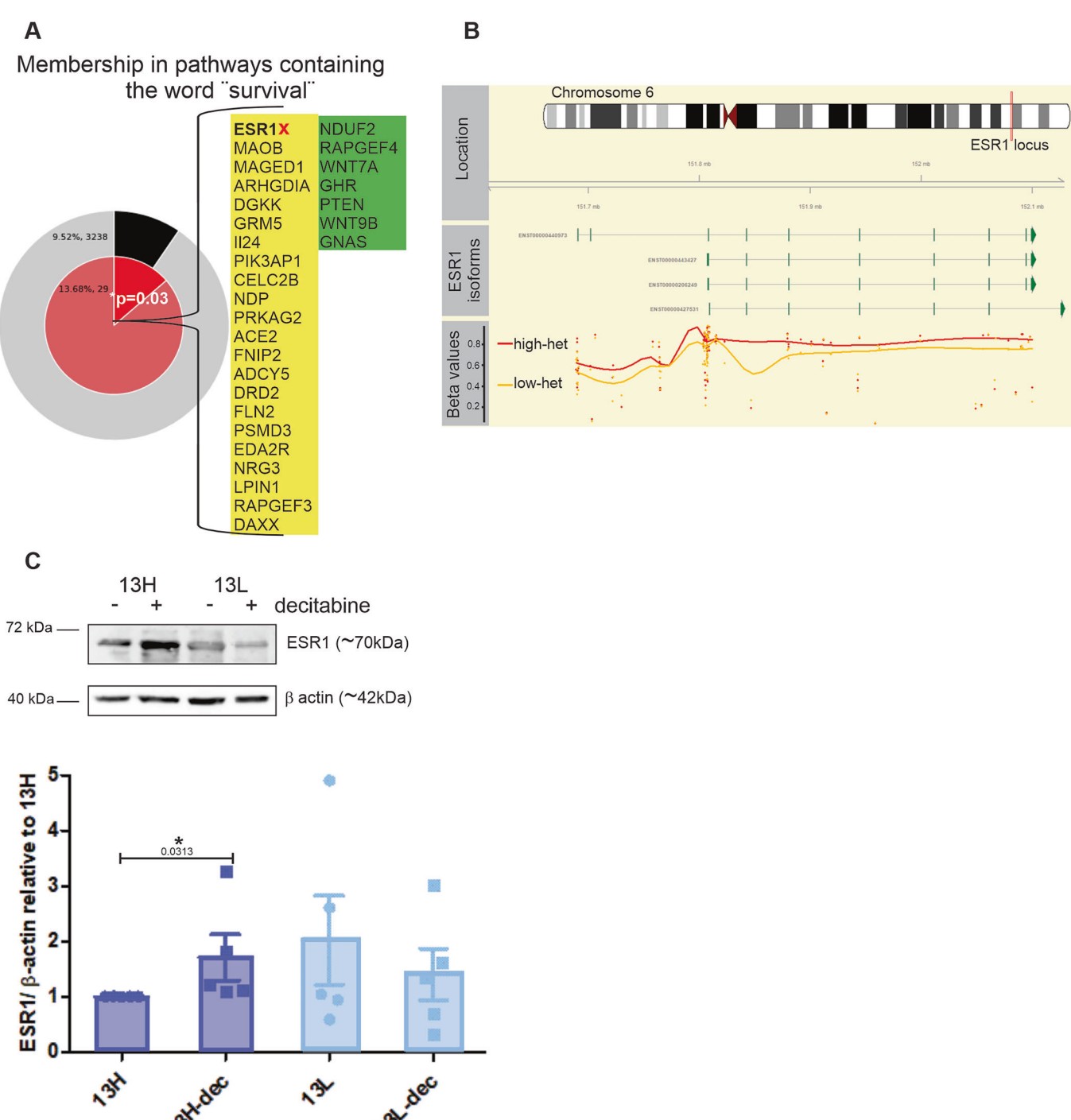

cells increased (Fig. 7B,C) and accordingly, the heteroplasmy of the mixture decreased (Fig. 7D).

## The DNA methylation inhibitor decitabine reduces mtDNA heteroplasmy in an in vivo xenograft model by erasing nuclear DNA methylation marks

After appreciating the effects of DNA methylation inhibitors on the particular heteroplasmy cybrids regarding cell fate and mutation load, we aimed to scale these observations to a more physiological in vivo

model. For this, we implanted subcutaneously 13H, 13L and a mixture of 13H and 13L cybrids in NSG mice. After tumors were palpable (~2 weeks after injection), we treated half the mice with decitabine and the other half with its vehicle (DMSO) for 8 days and monitored tumor growth. Post-mortem analysis of the tumors included nuclear and mitochondrial DNA methylation pattern assessment (to ensure dectiabine's in vivo action) and m.13513 G > A heteroplasmy (to test its potential therapeutic facet) (Fig. 8A). We found even more striking effects to what we had observed in cell culture. As expected, and as reported before (Osuna et al, 2019; Achinger-Kawecka et al, 2024;

**Figure 5. Genes related to survival are differentially methylated in high versus low heteroplasmy cybrids.**

(A) Metascape analysis of the differentially methylated genes that resulted from 850k EPIC (222 genes meeting differentially methylated criteria—see "Methods" section —of which 168 = 75.7% were hypermethylated), Dataset EV1. The Metascape pie chart compares membership in "survival pathway" of our gene list (inner pie) and the proportion of survival genes that would appear by chance in coding regions (outer pie). The significantly low $P$ value confirms that "survival" is overrepresented in our gene list, emphasizing its relevance to our hypothesis. The genes that determine the enrichment in survival pathways are listed next to the pie, in yellow are the hypermethylated and in green the hypomethylated genes for the high-heteroplasmy versus low-heteroplasmy group. Marked with a red cross is the ESR1 gene, which is analyzed further in (B, C). In Dataset EV1, metascape sheet, these genes are displayed next to the pathways to which they belong. (B) Differential DNA methylation of the ESR1 locus. The top panel represents the chromosomal location of ESR1. The middle panel depicts ESR1 transcript isoforms (ENSEMBL IDs), with exons as green boxes and introns as connecting lines. The bottom panel displays DNA methylation data from the Illumina Infinium MethylationEPIC array (850 K) for the ESR1 locus. Beta-values, which range from 0 (unmethylated) to 1 (fully methylated), for CpG probes in the region. Average beta-values for each group (high and low heteroplasmy) of the individual CpG sites are shown as dots. The red and gold lines are LOESS (locally estimated scatterplot smoothing) curves, that provide a smoothed representation of methylation differences across the region, confirming a higher methylation level for the high-heteroplasmy samples. (C) Representative western blot and quantification of ESR1 protein in the m.13513 G > A cybrids (13H= high-heteroplasmy, 13L= low-heteroplasmy) with and without 3-day 1 μM decitabine (dec) treatment. *$P < 0.05$, Wilcoxon matched-pairs signed rank test. $N = 5$ independent experiments. Bars represent mean $+/-$ SEM. Source data are available online for this figure.

Gutierrez et al, 2022), overall tumor growth was decelerated in the decitabine group (Fig. EV6A, left panel). When dissecting the different tumor types, the growth of high-heteroplasmy and "mixed" tumors was significantly attenuated with decitabine treatment, whereas the low-heteroplasmy ones did not respond to the demethylating agent (Fig. 8B). When comparing the growth rate of the purely high- or low-mutant tumors within the controls (vehicle-DMSO) and in the treated mice (decitabine) there were no significant differences. Although in the untreated group, the high-mutants tended to grow faster than the low-mutants, whereas the opposite was true for the treated group (Fig. EV6A, right panel), in line with what we had seen in cell culture. In conclusion, there is a distinctive tumor growth and sensitivity to DNA methylation modulation in relation to heteroplasmy that we thought could be exploited to shift heteroplasmy. Additional information regarding final tumor sizes is available in Fig. EV6B and its legend.

Regarding the drug´s impact on the tumors´ epigenome, as expected, decitabine reduced overall DNA methylation, and DNA methylation comparisons of high versus low-heteroplasmy tumors mirrored cultured cells´ EPIC results (Fig. 8C, left panel). The method we used to assess methyl-CpGs in this occasion (MinIOn Nanopore technology®) is especially useful for short genomes such as the mtDNA, so to the contrary of EPIC, mtDNA data was over-represented, and we confirmed the same observations reported with EPIC. As reported before (Aminuddin et al, 2020; Morris et al, 2018), mtDNA methylation was minimal compared to the nuclear one and importantly, decitabine had no significant effect on this pattern (Fig. 8C, right panel, Fig. EV6C). Mean methylation levels across the mitochondrial chromosome notably showed an opposite trend (higher methylation for the treated tumors) (Figs. 8C and EV6C). Therefore, we can confirm that the high-low-heteroplasmy differential behavior is determined through nuclear changes and that decitabine´s effect is centered in nuclear epigenetic modification.

Finally, and most importantly, we checked the impact of decitabine on heteroplasmy of all tumors. Again, as we had seen in the cell culture experiments, decitabine reduced the mutation load significantly in the high-heteroplasmic tumors and the mixed heteroplasmy ones. Heteroplasmy from the low group was not modified by the demethylating drug (Figs. 8D and EV6D). Notably, the heteroplasmy reduction in the in vivo model was more pronounced than in cell culture, achieving a 22% reduction (7% more than in cells) for the high-heteroplasmy tumors and a 40% reduction, (doubling the effect in cells) for the mixed heteroplasmy ones.

## Discussion

Our study provides new insights into how mitochondrial health influences nuclear epigenetics (Mayorga et al, 2019; Kopinski et al, 2019), and how nuclear DNA methylation in turn, impacts mitochondrial function.

Specifically, we show that both the type (mtDNA mutation) and intensity (heteroplasmy) of mitochondrial dysfunction distinctly shape nuclear epigenetics. In line with our previous findings, we confirmed that severe mitochondrial impairment correlates with increased nDNA methylation (Mayorga et al, 2019). This epigenetic patterning is especially important for the survival of high-heteroplasmy cells, supporting their proliferation and helping them avoid apoptosis. This aligns with previous findings in 143B osteosarcoma cells, which share the nuclear background of our cybrid model, showing that impaired respiration can paradoxically protect cells from apoptosis (Dey and Moraes, 2000). Consistently, disrupting nuclear methylation selectively compromised high-heteroplasmy cells, as demonstrated by our cell culture and xenograft model, where decitabine treatment altered nuclear but not mitochondrial DNA methylation. While mtDNA methylation remains debated (Morris et al, 2018; Stoccoro and Coppedè, 2021), we observed only minor levels in our cell lines with no significant differences across comparisons, reinforcing that major epigenetic shifts triggered by mitochondrial dysfunction occur in nuclear DNA. These findings emphasize the pivotal role of mitochondria-to-nucleus communication for the epigenetic adaptation in a mitochondrial dysfunctional scenario.

After confirming that mitochondrial function molds the nuclear epigenome, we demonstrated that, conversely, nDNA methylation modulation has an impact on mitochondrial function, and most interestingly on mtDNA heteroplasmy. Our studies showed that FDA-approved DNA methylation inhibitors, particularly decitabine, had an impact on mitochondrial respiration and reactive oxygen species production. Although we did not observe changes in the expression levels of mitochondrial complex proteins—having assessed only five subunits—the possibility remains that ATP synthase (Complex V) activity could be enhanced by demethylating agents (Yang et al, 2016), potentially accounting for the observed increases in ATP production, mild uncoupling, elevated oxygen consumption rate (OCR) and ROS levels. Furthermore, the loss of the adaptive, methylation-dependent mechanisms in cells with high heteroplasmy may render them more susceptible to metabolic

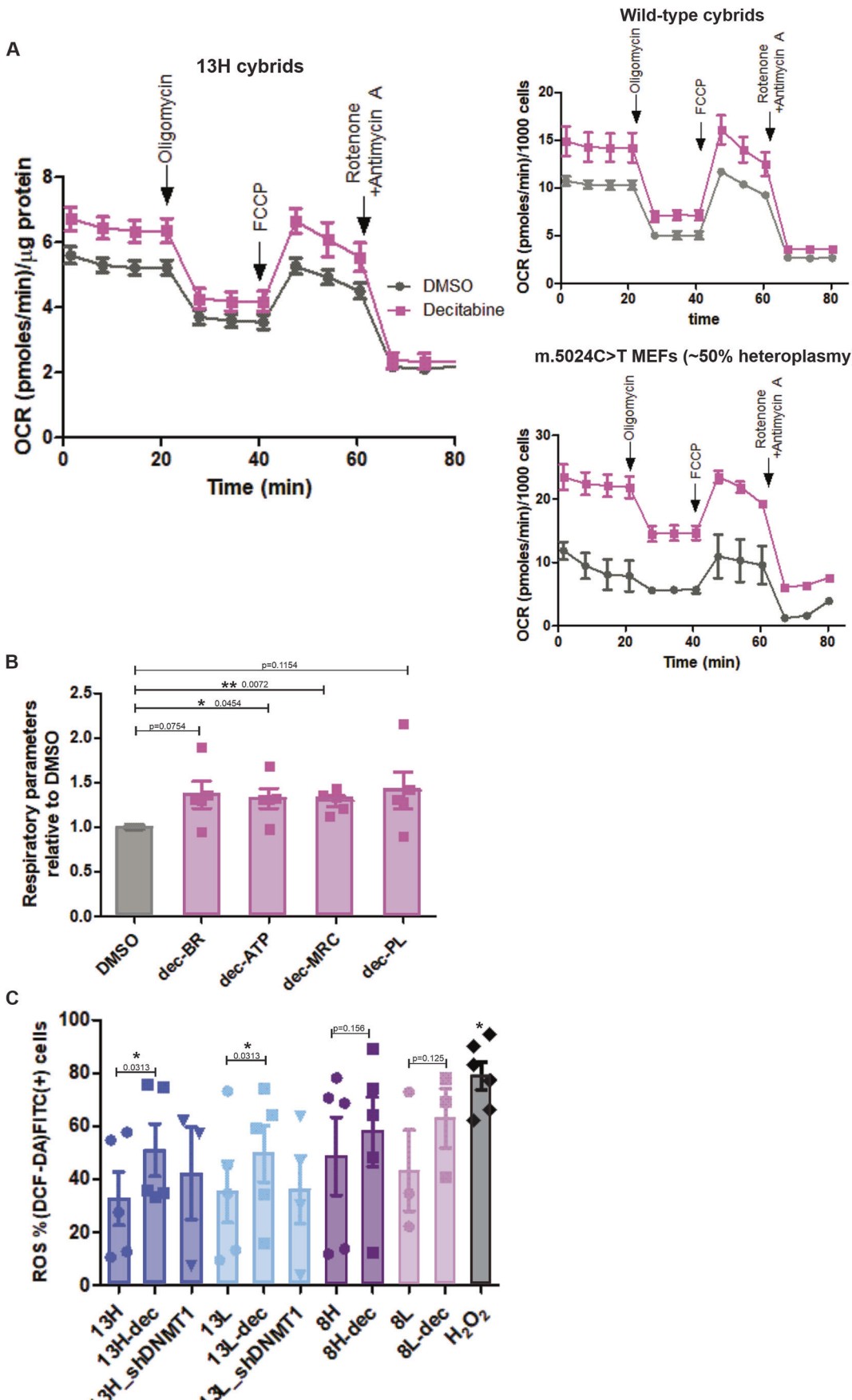

◀ **Figure 6. DNA methylation inhibitor decitabine impacts mitochondrial function.**

(A, B) High-throughput oxygen flux analysis using Seahorse®. Cells (left panel: 13H, right panel wild-type cybrids and m.5024 C > T ~ 50% heteroplasmy MEFs) were submitted to a mito-stress test. (A) Oxygen consumption rate (OCR) kinetics with sequential addition of Oligomycin, FCCP, Rotenone+ Antimycin A. Values were normalized to protein amount or cell number. (B) Quantification of key mitochondrial function parameters of cells treated with decitabine (dec) were relativized to control DMSO condition. BR = Basal respiration (OCR at baseline- OCR post_Rot+Antimycin), ATP = ATP production (OCR Basal respiration − OCR post_oligomycin), MRC= Maximal respiration capacity (OCR post_FCCP − OCR post_Rot+Antimycin), PL= proton leakage (OCR post-Oligomycin—OCR post_Rot+Antimycin). ATP and MRC increased in the decitabine (dec) conditions 1 μM 1-day treatment. Basal respiration and proton leakage showed a tendency to do the same **$P < 0.01$, *$P < 0.05$. Two-tailed, one sample Student´s $T$ test comparing to DMSO-control. $N = 5$ independent experiments. Three 13H, one wild-type cybrids and one m.5024 C > T MEFs. (C) Evaluation of reactive oxygen species (ROS) through flow cytometry using DCF-DA (FITC): % of FITC(+) cells were determined using an autofluorescence-based threshold. % of cells testing positive for ROS were compared using one-tailed, paired Wilcoxon matched-pairs signed rank test. *$P < 0.05$. $N = 3$–5 independent biological experiments, comparisons between treated and untreated conditions were paired within each experiment. Decitabine (dec) 1 μM 3-day treatment versus controls-DMSO. Positive control: $H_2O_2$, *$P < 0.05$, significance is shown for comparison to the control condition with the highest FITC(+) mean(8H), one-tailed unpaired Student´s $t$ test. Bars represent mean +/− SEM. Source data are available online for this figure.

imbalance, explaining the distinct proliferation and apoptosis responses observed in relation to heteroplasmy levels.

Building on these insights, we visioned that the selective vulnerability of high-mutant heteroplasmic cells towards DNA methylation inhibitors could be a tool to lower heteroplasmy and proved their capability to do so. By targeting these methylation patterns with drugs like decitabine and 5-azacytidine, the growth and survival of high-heteroplasmy cells was impaired, resulting in the enrichment of low-heteroplasmy cybrids, decreasing the overall mutant load. This was shown both in cell culture and most pronouncedly in an in vivo xenograft model.

This study introduces a groundbreaking approach to addressing mitochondrial heteroplasmy by modulating nuclear epigenetics rather than directly targeting mtDNA. This proof of concept opens a new window for epigenetic modifiers in the treatment of mitochondrial disorders. Unlike other methods that require delivery of foreign molecules to the mitochondria (Hashimoto et al, 2015; Bacman et al, 2013; Gammage et al, 2014; Bayona-Bafaluy et al, 2005; Srivastava and Moraes, 2001; Mok et al, 2020), our strategy exploits endogenous nuclear-mitochondrial communication pathways. Furthermore, this approach avoids transient mtDNA depletion—a common challenge with nuclease-based therapies (Bacman et al, 2013; Hashimoto et al, 2015; Shoop et al, 2023)—and relies on shifting the cellular population dynamics to favor healthier mitochondrial compositions. The use of clinically approved drugs may also facilitate the translation of this strategy into clinical practice.

While promising, our findings come with limitations. The use of cybrid cell lines and xenograft models, though powerful, may not fully capture the complexity of tissues most affected by mitochondrial diseases, particularly post-mitotic ones with minimal cell turnover. On the one hand, DNA methylation inhibitors require proliferating cells to exert their effect (Foulks et al, 2012; Stresemann and Lyko, 2008; Christman, 2002). On the other hand, the selective elimination of high-heteroplasmy cells may have functional consequences that warrant further exploration. The increasing availability of mtDNA disease animal models (Khotina et al, 2023; Guo et al, 2021; Lee et al, 2022; Sabharwal et al, 2021) certainly compels additional in vivo research to test the possibility to lower heteroplasmy and alleviate symptoms with epigenetic drugs in these settings.

Investigating combination therapies—such as pairing DNA methylation inhibitors with mitochondrial-targeted antioxidants (Barcelos et al, 2020; Jiang et al, 2020), or post-mitotic tissue regenerating strategies (Kadi et al, 2004; Vidman et al, 2025; Fine and Vunjak-Novakovic, 2020; Murphy et al, 2008)—could further

enhance therapeutic outcomes by increasing the drugs´ action and mitigating potential side effects.

Moreover, beyond mitochondrial diseases, the susceptibility of cells with very poor mitochondrial function to epigenetic modulation holds potential for applications in oncology. Although DNA methylation modulators are already used in this field (Gawlitza et al, 2019; Christman, 2002), these drugs could be specifically efficient in cancers with particularly unfit mitochondria (Luo et al, 2020), broadening the clinical utility of the findings of our work.

In conclusion, our study identifies a DNA methylation-dependent adaptive survival mechanism in high-heteroplasmy cells, which can be selectively targeted to reduce mtDNA heteroplasmy. This work introduces epigenetic modulation as a novel strategy to shift mitochondrial mutant load, offering a promising therapeutic avenue for mitochondrial diseases and potentially other conditions involving mitochondrial dysfunction.

# Methods

**Reagents and tools table**

| Reagent/resource | Reference or source | Identifier or catalog number |
|---|---|---|
| **Experimental models** | | |
| Cybrids= fusion of cell line143B.206 ρ0 with enucleated patient-derived dermal fibroblasts | Bacman et al, 2020; Hashimoto et al, 2015 Moraes lab, University of Miami | N/A |
| Mouse embryonic fibroblasts (MEFs) derived from a heteroplasmic mouse model carrying the mtDNA tRNA-Ala m.5024 C > T mutation | Kauppila et al, 2016, Moraes lab, University of Miami | N/A |
| Nod Scid Gamma (NSG) mice (NOD.Cg-PrkdcscidIl2rgtm1Wjl/SzJ, NSG) | The Jackson Laboratory | RRID:IMSR_JAX:005557 |
| **Recombinant DNA** | | |
| pLV[shRNA]-LacI: T2A:Puro-U6/2xLacO > hDNMT1[shRNA#1] | Vector builder | N/A |
| MISSION® Lentiviral Packaging Mix | Sigma-Aldrich | SHP001 |

| Reagent/resource | Reference or source | Identifier or catalog number |
|---|---|---|
| **Antibodies** | | |
| Anti-DNMT1 | antibodies.com | A307589 |
| Anti-ESR1 | Cusabio | CSB-PA11399A0Rb |
| Total OXPHOS rodent WB antibody cocktail | abcam | ab110413 |
| Anti-β-actin, clone 15G5A11/E2 | Invitrogen | MA1-140 |
| Anti-vinculin, clone V284 | Sigma-Aldrich | CP74 |
| Anti-vinculin | Assay Genie | CAB2752 |
| Gt anti-mouse IgG -H + L | Invitrogen | 31430 |
| Gt anti-Rb IgG -H + L | Invitrogen | 31460 |
| Anti-MAP1 LC3B | antibodies.com | A7198 |
| Anti-TOMM20, clone AT1B2 | LSBio | LS-C755581 |
| Goat Anti-Rabbit IgG H&L (Alexa Fluor® 488) | Abcam | ab150077 |
| Goat Anti-Mouse IgG H&L (Alexa Fluor® 647) | Abcam | ab150115 |
| **Oligonucleotides and other sequence-based reagents** | | |
| PCR primers | This study (see Table in "Methods") | N/A |
| **Chemicals, enzymes, and other reagents** | | |
| Ethidium bromide | Sigma-Aldrich | E7637 |
| DMEM high glucose | Gibco | 11995065 |
| Fetal bovine serum | Gibco | 16000044 |
| Penicilin+ Streptomicine | Gibco | 15140122 |
| Uridine | Sigma-Aldrich | U3003 |
| Tris | Bio-Rad | 1610719 |
| EDTA | Bio-Rad | 1610729 |
| CTAB | Sigma-Aldrich | H6269 |
| 2-mercaptoethanol | Bio-Rad | 1610710 |
| chloroform | Sigma-Aldrich | C7559 |
| HCl | Merck Millipore | K46497817513 |
| 2-propanol | Sintogran | SIN-026003-01 |
| Ethanol | Sintogran | SIN-132003-01 |
| DNA PuriPrep-S kit | Inbio Highway | K1205-50 |
| SsoAdv univer SYBR GRN | Bio-Rad | 1725271 |
| MS-MLPA salsa probemixes | MRC-Holland | ME001-C2, ME001-D1, ME002-C2 |
| SALSA MLPA Reagent Kit | MRC-Holland | EK1-FAM |
| Cfol | Promega | R624A |
| EZ DNA Methylation—GoldTM kit | ZymoResearch | D5006 |
| Infinium MethylationEPIC BeadChipv.1.0 | Illumina | WG-317-1001 |
| Rapid Barcoding Kit 24 V14 | Oxford Nanopore Technologies | SQK-RBK114.24 |
| 5-azacytidine | Sigma-Aldrich | A2385 |

| Reagent/resource | Reference or source | Identifier or catalog number |
|---|---|---|
| 5-aza-2′-deoxycytidine= decitabine | Sigma-Aldrich | 189826 |
| DMSO | Sigma-Aldrich | D2650 |
| Polyethylenimine(PEI) | Poliscience® | 24765 |
| 0.22-mm filter, cellulose acetate | Sartorius Minisart | 1875118 |
| Polybrene | Sigma-Aldrich | TR-1003 |
| Puromycin | Santa Cruz Biotechnology | sc-108071 |
| IPTG: isopropyl-galactosidase | TransGen Biotech | GF101-01 |
| Proliferation Assay Kit | Assay Genie | BN00566 |
| Tetramethylrhodamine ethyl ester, perchlorate (TMRE) | Molecular probes® | T669 |
| Carbonylcyanide m-chlorophenyl hydrazone (CCCP) | Sigma-Aldrich | C2759 |
| Annexin V-FITC | BD Biosciences | BDB-556420 |
| ANNEXIN V binding buffer | BD Biosciences | BDB-556454 |
| TRITON | Biopack | 2000200207 |
| NaCl | Cicarelli | 750814 |
| Protease inhibitor cocktail Halt | ThermoFisher | 87786 |
| Pierce™ BCA Protein Assay Kit | ThermoFisher | 23227 |
| TGX Stain-free gels | Bio-Rad | 456-8096 |
| 30% Acrylamide/Bis Solution, 29:1 | Bio-Rad | 1610156 |
| SDS | Genbiotech | RU2407 |
| Ammonium persulfate | Bio-Rad | 1610700 |
| TEMED | ThermoFisher | 17919 |
| Western ECL | BPSBioscience | 79572-1 |
| Oligomycin | Sigma-Aldrich | 75351 |
| Trifluoromethoxy carbonylcyanide phenylhydrazone (FCCP) | Sigma-Aldrich | C2920 |
| Rotenone | Sigma-Aldrich | R8875 |
| Antimycin A | Sigma-Aldrich | A8674 |
| Detergent Compatible (DC) Protein Assay | Bio-Rad | 5000121 |
| Quinacrine dihydrochloride | Abcam | ab145375 |
| 2′,7′- Dichlorofluorescin diacetate (DCF-DA) | Sigma-Aldrich® | D6883 |
| $H_2O_2$ | Sigma-Aldrich® | HX0640 |
| 5-Chloromethylfluorescein Diacetate (CMFDA) | abcam® | 122297 |
| 4% Paraformaldehyde | Sigma-Aldrich® | 158127 |
| ClNH$_4$ | Biopack | 2000167906 |
| Albumin 2%/Saponin0.1% | Biopack | 2000946003 |
| Hoechst 34580 | Thermo Scientific® | 62249 |

| Reagent/resource | Reference or source | Identifier or catalog number |
|---|---|---|
| Mowiol | Sigma-Aldrich | 475904-M |
| Polyethylenimine(PEI) | Poliscience® | 24765 |
| **Software** | | |
| GeneMarker v1.75 software | Softgenetics | GM001 |
| Illumina iScan system | Illumina | N/A |
| GenomeStudio, Methylation module | Illumina | N/A |
| GViz R package v1.46.1 | N/A | N/A |
| Dorado basecaller v0.9.1 | Oxford Nanopore Technologies | N/A |
| Samtools | N/A | N/A |
| ModKit v0.4.4 | Oxford Nanopore Technologies | N/A |
| NanoMethViz v2.8.1 | Su et al, 2021 | N/A |
| Rsamtools v2.18.0 | N/A | N/A |
| FlowJo v X.0.7 | BD Biosciences | N/A |
| ImageJ | N/A | N/A |
| GraphPad Prism v5.03 | GraphPad Software, LLC | N/A |
| Corel draw 2021 | Corel Corporation | N/A |
| **Other** | | |
| Aria Mx-Real time PCR system | Agilent technologies | N/A |
| ABI-3130 sequencer | Applied Biosystems | N/A |
| Metascape | Zhou et al, 2019 | N/A |
| MinION Mk1C | Oxford Nanopore Technologies | N/A |
| Fluoroskan microplate reader | Thermo Scientific | N/A |
| FACSAria flow cytometer | BD Biosciences | N/A |
| LAS Fujifilm 4000 system | GE Healthcare Life Sciences | N/A |
| Seahorse XFp Extracellular Flux Analyzer | Agilent technologies | S7802A |
| TC20 Automated Cell Counter | Bio-Rad | 1450102 |
| Fluorescence confocal microscopy | Olympus Confocal Microscope FV1000-EVA | N/A |

## Methods and protocols

### Mitochondrial disease model

**Cybrids** were obtained from the fusion of the osteosarcoma cell line143B.206 ρ0 mtDNA devoid cells (human-female) with enucleated patient-derived dermal fibroblasts as previously described (Bacman et al, 2020; Hashimoto et al, 2015). Informed consent was obtained from all human subjects and experiments conformed to the principles set out in the WMA Declaration of Helsinki and the Department of Health and Human Services Belmont Report. We used cybrids harboring wild-type mtDNA and

two disease-causing mutations: m.13513 G > A (MT-ND5) (Shanske et al, 2008; Wang et al, 2010, 2023) and m.8344 A > G (MT-tRNA-Lys) (Silvestri et al, 1993; Shoffner et al, 1990). Clones with different levels of heteroplasmy were obtained by treating cybrids with Ethidium Bromide, Sigma-Aldrich cat#E7637 (50 ng/ml—2 weeks) to partially deplete them from mtDNA followed by cell dilution, clone selection and random mtDNA repopulation (Bacman et al, 2020) in Ethidium Bromide-free medium. High (above the usual disease-triggering threshold) and low (below the threshold) heteroplasmy clones were selected for further experiments. The clones were abbreviated as 13H (cybrids harboring ≥70% m.13513 G > A mutation), 13L (≤30% m.13513 G > A mutation), 8H (≥95% m.8344 A > G mutation), 8L (<50% m.8344 A > G mutation, although for cell culture experiments 8L with <5% heteroplasmy were used).

For a few additional experiments, we also used embryonic fibroblasts (MEFs) derived from a heteroplasmic mouse model carrying the mtDNA tRNA-Ala m.5024 C > T mutation (Kauppila et al, 2016) which were immortalized with the E6-E7 gene from the human papilloma virus (HPV) (Lochmüller et al, 1999) and human dermal fibroblasts containing the m.14459 G > A MT-ND6 mutation (also immortalized with the HPV E6-E7 gene).

Cells were grown at 37 °C, 5% $CO_2$, in DMEM high glucose (Gibco®, cat#11995065), supplemented with 10% fetal bovine serum (Gibco®, cat#16000044), 100U/ml Penicilin+ Streptomicine (Gibco®, cat#15140122) and uridine (Sigma-Aldrich ®U3003) 50 μg/ml.

### DNA extraction

CTAB/chloroform-isoamylic based protocol: Cells were collected and tissues were homogenized in PBS using an Ultra turrax homogenizer, following centrifugation, the pellet (cells or tissue) was suspended and washed one time with Tris–EDTA ($T_{10}E_{10}$), suspended in cetyltrimethylammonium bromide (CTAB) solution (2 g/l CTAB Sigma-Aldrich®, 100 mM Tris/HCl, 20 mM EDTA and 2% 2-mercaptoethanol) and incubated at 60 °C for 1 h for cells or overnight for tissue. Then, chloroform-isoamyl alcohol solution (24:1) was added, and the sample was centrifuged. The aqueous phase was collected into a new tube and mixed with 3 volumes of ice-cold 100% ethanol. Precipitated DNA was dissolved in $T_{10}E_{0.1}$.

Alternatively, when few cells were available, DNA was purified using the DNA PuriPrep-S kit (Inbio Highway®, cat#K1205-50), following the manufacturer´s protocol.

### Heteroplasmy and mtDNA content assessment (SYBR-based qPCR)

qPCR was carried out using SsoAdv univer SYBR GRN Bio-Rad® (cat#1725271). In total, 1 ng of total DNA was used for each PCR reaction. Aria Mx-Real time PCR system/Agylent technologies®) was used to readout the PCRs.

*mtDNA mutation heteroplasmy* was measured using three primers for each mutation: one forward primer (F) and two reverse primers: one specific for the mutated version (R-mut) and the other a few base pairs downstream to the mutation with the ability to amplify both mutated and wild-type molecules (R-all). Heteroplasmy was quantified using ΔCT method= $2^{-(mutCT-allCT)}$.

*mtDNA content* was estimated by normalizing the mtDNA "all" amplicon to the one of a nuclear gene (B2M), quantified using ΔCT method= $2^{-(allCT-B2MCT)}$.

Primers

| Mutation or gene | Strand | Sequence 5´-3´ |
|---|---|---|
| m.13513 G > A (homo sapiens) | F | GGGTCCATCATCCACAACCTT |
| | R-mut | GCGGTTTCGATGATGTGGTT |
| | R-all | TTGCGGTTTCGATGATGTGG |
| m.8344 A > G (homo sapiens) | F | AAACCACAGTTTCATGCCC |
| | R-mut | CACTGTAAAGAGGTGTTGGC |
| | R-all | TTCACTGTAAAGAGGTGTTGG |
| B2M (homo sapiens) | F | GACTTGTCTTTCAGCAAGGA |
| | R | ACAAAGTCACATGGTTCACA |
| m.5024 C > T (mus musculus) | F | AGCTATCATAAGCACAATAACCC |
| | R-mut | AATAGATGTAGGATGAAGTCTTACGA |
| | R-all | CAATAGATGTAGGATGAAGTCTTACA |

qPCR conditions

| Mutation or gene | Amplicon | Primer concentration (F and R) | Thermal cycling conditions |
|---|---|---|---|
| m.13513 G > A | 13-mut | 0.25 µM | Start: 95 °C 3′ Cycling (×40): 95 °C 15″ 60 °C 30″ (reading) 72 °C 30″ |
| | 13-all | 0.25 µM | |
| m.8344 A > G | 8-mut | 0.25 µM | Start: 95 °C 3′ Cycling (×40): 95 °C 15″ 60 °C 30″ (reading) 72 °C 30″ |
| | 8-all | 0.5 µM | |
| B2M | B2M | 0.25 µM | Start: 95 °C 3′ Cycling (×40): 95 °C 15″ 60 °C 30″ (reading) 72 °C 30″ |
| m.5024 C > T | 5024-mut | 0.15 µM | Start: 95 °C 3′ Cycling (×40): 95 °C 15″ 58 °C 30″ (reading) 72 °C 30″ |
| | 5024-all | 0.5 µM | |

### DNA methylation studies

Methyl-specific-multiplex ligation-dependent probe amplification (MS-MLPA) (Nygren et al, 2005): The ME001-C2 or D1 and ME002-C2 kits (MRC-Holland®) were utilized for the assays. The MS-MLPA process involved the following steps: 50–100 ng of total DNA was used to initiate the reaction, following adding probes which hybridized for 18 h at 60 °C. Subsequently, ligation was carried out at 54 °C, and the reaction mixture was split into two halves. One-half was digested at 37 °C for 30 min with the methyl-sensitive restriction enzyme CfoI (Promega® R624A), which specifically cleaves unmethylated 5′-GCGC-3′ sequences, while the other half was left undigested to serve as a reference. Both ligated samples were then amplified via PCR using FAM-labeled MLPA primers. The resulting fluorescent-labeled PCR products were analyzed by capillary electrophoresis on an ABI-3130

sequencer (Applied Biosystems®) and further processed with GeneMarker v1.75 software (Softgenetics®). The mean methylation value for each CpG site from each condition was used for hierarchical clustering and heatmap confection.

Methylation analysis using Infinium EPIC 850k chips (Pidsley et al, 2016) (Dataset EV1, and (GEO) GSE300902): ~500 ng genomic DNA (from 14 samples: two wild-type, three 13H, three 13L, three 8H, three 8L samples) was submitted to bisulfite conversion using the EZ DNA Methylation—GoldTM kit (ZymoResearch®). Bisulfite converted DNA was thereafter hybridized to the Illumina Infinium MethylationEPIC BeadChip, representing the methylation state of over 850,000 CpG sites. The array was imaged using the Illumina iScan system (Illumina®). GenomeStudio® software was used for processing. Steps as follows, Quality check → Background Correction & Dye Bias Equalization → Filtering → BMIQ Normalization → Data Transformation. Each methylation data point is represented by fluorescent signals from the M (methylated) and U (unmethylated) alleles. Background intensity computed from a set of negative controls was subtracted from each analytical data point. The ratio of fluorescent signals was then computed from the two alleles Beta value = $(max(M, 0))/(|U| + |M| + 100)$. The Beta-value reflects the methylation level of each CpG site. A Beta-value of 0–1 was reported signifying percent methylation, from 0% to 100%, respectively. To calculate differential methylation for each CpG site, we used the following formula:

$\Delta$ (delta) = mean(Avg_beta of Test group) - mean(Avg_beta of Control group).

We performed mutated vs. wild-type and high vs. low-heteroplasmy comparisons.

Beta-values from each site and sample are available in Dataset EV1.

Identification of significantly differentially methylated CpG sites:

CpG sites were considered hypermethylated if $\Delta \geq 0.2$, and hypomethylated if $\Delta \leq -0.2$.

In addition, significance required a $P$ value < 0.05 from a one-tailed, unpaired Student's $t$ test, and the differential methylation needed to occur towards the same direction (hyper or hypo) in two or more consecutive CpG sites in the genome (based on the Infinium MethylationEPIC v1.0 B5 Manifest File).

From these CpG sites, we selected the ones associated to particular genes and that list was imported to the Metascape online tool (http://metascape.org) (Zhou et al, 2019) to perform "custom" analysis, searching for enrichment of genes under membership categories in any pathway that involved the term survival (in pathway name or description).

Regarding ESR1 associated CpG sites, we ploted their beta-values using GViz R package v1.46.1 and dependencies. The scripts used for data processing are available in our GitHub repository https://github.com/slaurito/Methylation-Analysis-from-Nanopore-and-Epic-Data-Perez-et-al.

Methylation analysis and mtDNA sequencing using (MinION) Oxford Nanopore Technologies® (Clarke et al, 2009) (Dataset EV2 and Tabix file (available at https://www.mediafire.com/file/q9xjh6it3t0cu7s/Tabix.tsv.bgz/file)): ~200 ng DNA samples from xenograft tumors conditions (13H, 13H-dec, 13L and 13L-dec, three tumors from each condition, total of 12 samples) were used as input for Nanopore sequencing. The library was prepared using the Rapid Barcoding Kit 24 V14 (SQK-RBK114.24) from Oxford Nanopore Technologies, following the manufacturer's instructions.

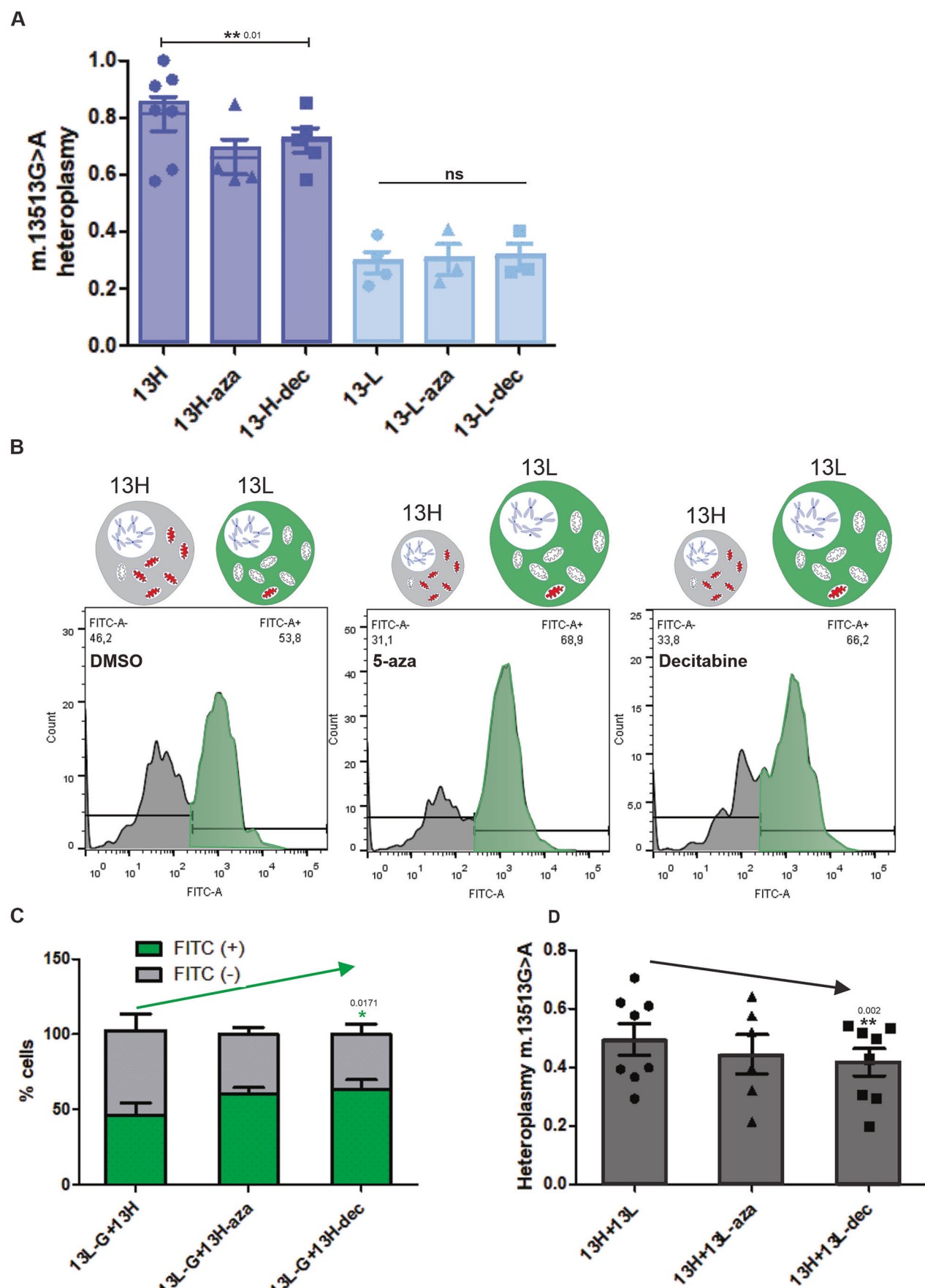

**Figure 7. Treatment with DNA methylation modulators lower m.13513 G > A cybridś heteroplasmy.**

(A) m.13513 G > A heteroplasmy was measured using SYBR green-based RT-qPCR. M.13513 G > A high-heteroplasmy cells (13H) decreased their heteroplasmy when treated with decitabine (dec) 1 μM—3 days. One-tailed, paired (13H vs 13H-aza; 13H vs 13H-dec comparisons) Student´s $t$ test. $N = 5$ independent experiments with decitabine, $N = 4$ with 5-azacytidine (aza) that showed the same tendency ($P = 0.125$). (B) Low-heteroplasmy m.13513 G > A cybrids traced with green cell tracker (13L-G) were mixed in equal proportions with undyed high-heteroplasmy cybrids (13H) before treating them with DMSO, 5-azacytidine(aza) or decitabine(dec). After 3-day treatment they were collected and the percentage of green cells= %FITC(+)= (13L-G) was measured through flow cytometry. (C) Quantification of the percentage of FITC (+) = 13L-G and FITC (−) = 13H cells after treating them with DMSO, azacytidine or decitabine. One-tailed, paired Student´s $t$ test comparing the % of FITC(+) cells between control and treated cells *$P < 0.05$. $N = 4$ independent experiments. (D) m.13513 G > A heteroplasmy of the mixture of 13H + 13L cells treated with DMSO, 5-azacytidine or decitabine was measured using SYBR green-based RT-qPCR. **$P < 0.01$ One-tailed, paired Student´s $t$ test (control versus aza and control versus decitabine). Treatment with 5-azacytidine showed the same tendency ($P = 0.0708$). $N = 8$ independent experiments with control-decitabine, 6 with control-5-azacytidine conditions. Bars represent mean $+/−$ SEM. Source data are available online for this figure.

Briefly, this system employs transposome complexes that cleave the DNA and add barcodes to each sample. The samples are then pooled, and the library is purified using magnetic beads. Finally, sequencing-specific adapters are added. Sequencing was performed on a MinION Mk1C device for 48 h at 37 °C, generating POD5 files for subsequent analysis. POD5 files were processed for base calling using Dorado basecaller v0.9.1, including the identification of 5-methylcytosine and 5-hydroxymethylcytosine using the high-accuracy model (HAC). The resulting. BAM files were aligned to the reference genome Hg38, obtained from the UCSC Genome Browser (https://genome.ucsc.edu/), using Dorado aligner. Subsequently, the. BAM files were sorted and indexed using SamTools. In addition, for further analyses, bedmethyl files containing the genomic coordinates of all modified cytosines were obtained using ModKit v0.4.4.For the bioinformatics analysis, the NanoMethViz (Su et al, 2021) v2.8.1 package was initially used to generate a Tabix file (available at https://www.mediafire.com/file/q9xjh6it3t0cu7s/Tabix.tsv.bgz/file) containing the chromosomal coordinates of each detected cytosine modification and its Log-Likelihood Ratio (LLR), which indicates the probability of cytosine modification. The functions NanoMethResult() and plot_region() were employed to generate heatmaps and visualize methylation levels across the mitochondrial genome. For global methylation analysis at the chromosomal and total genome levels, the methy_to_bsseq() and bsseq_to_log_methy_ratio() functions were used to convert the LLR of each genomic position into a logarithmic methylation ratio (LMR). Prior to statistical analysis and plot generation, the log-methylation ratios were transformed into values ranging from 0 (unmethylated) to 1 (methylated) using a sigmoidal function: $1/1+e^{-LMR}$. Statistical analyses were performed using the Shapiro–Wilk test for normality assessment and then the Wilcoxon test.

For the analysis of the mitochondrial genome mutation 13513 G > A, Rsamtools v2.18.0 was used. Heteroplasmy was calculated as ´A´ counts/(´A´counts + ´G´ counts).

The scripts used for data processing are available in our GitHub repository https://github.com/slaurito/Methylation-Analysis-from-Nanopore-and-Epic-Data-Perez-et-al.

### DNA methylation modulation treatments in cell culture

DNA methylation inhibition through drugs: Equal amounts of cybrid cells were plated (separately or in a 1:1 high:low-heteroplasmy mixture) to a 70% confluency for 1-day treatments or to a 50% confluency for 3-day treatments. 24 h afterwards, 5-azacytidine (Sigma-Aldrich®, cat#A2385) or 5-aza-2′-deoxycytidine= decitabine (Sigma-Aldrich® cat#189826) was added at different concentrations, i.e., 0.1 μM, 1 μM and 10 μM. Equal amounts of their vehicle (DMSO, Sigma-Aldrich® cat# D2650) were used for control conditions. For 3-day treatments, drugs were replenished every 24 h.

DNA methylation inhibition through IPTG-Inducible shRNA DNMT1 knockdown lentiviral vector: Generation of a stable cell line of m.13513 G > A low and high-heteroplasmy cybrids with the mammalian pLV[shRNA]-LacI: T2A:Puro-U6/2xLacO > hDNMT1[shRNA#1] purchased in Vector Builder®. Lentivirus were produced in Hek293T cells by transient co-transfection of the MISSION® Lentiviral Packaging Mix (Sigma®, cat#SHP001) with 1.35 μg of the shRNA-encoding DNA using 7 μg polyethylenimine (PEI, Poliscience®) for transfection of a well from a six-well plate. Culture supernatants were collected 48 and 72 h post transfection, filtered (0.22 mm filter, cellulose acetate, Ministart®), and supplemented with 8 μg/ml polybrene (Sigma®, cat#24765). Subconfluent cybrids (~70%) in a well from a six-well plate were infected with the polybrene-supplemented supernatant and 48 h post-infection were selected with 1 μg/ml puromycin (sc-108071®) for 1 week. These cybrids were treated with IPTG: isopropyl-galactosidase TransGen Biotech®cat# GF101-01 500 μM for 72 h (replacing medium and IPTG every day) and DNMT1 down-regulation was checked through western blot (Fig. EV2).

### Proliferation assays

We used Cell Proliferation Assay Kit (Assay Genie® BN00566): ~3000 cells/well were plated in a 96-well plate before treatment with demethylating agents. 24 h afterwards, basal cell density was measured with fluorometer (Fluoroskan™ Thermo Scientific™ microplate reader) after following proliferation assay´s protocol and treatments with drugs were started. At day 3, proliferation assay and fluorescence measurement were repeated as in the basal condition. For quantification, fluorescence after 3-day treatment was relativized to basal fluorescence before treatments.

### Mitochondrial membrane potential analysis

Tetramethylrhodamine ethyl ester, perchlorate (TMRE) (Molecular probes®, cat#T669) was used. Cells were incubated with 50 nM TMRE at 37 °C in culture medium in the dark for 30 min; cells were then taken up and passed through flow cytometer (FACSAria flow cytometer_BD®). The fluorescence was evaluated using excitation at 488 nm and read with band pass filter 585/42 nm. As a positive depolarization control, cells were treated with 100 μM Carbonyl-cyanide m-chlorophenyl hydrazone (CCCP, Sigma-Aldrich® cat#C2759) concomitantly with TMRE. Results were analyzed with FlowJo v X.0.7® software.

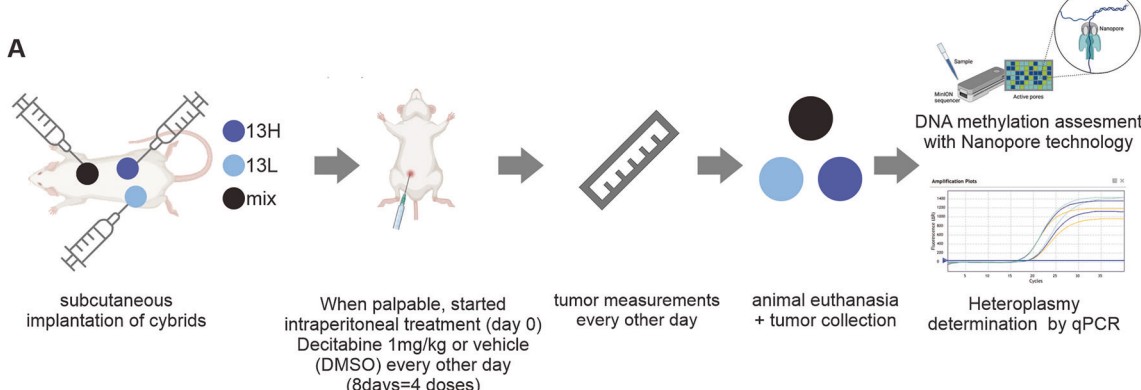

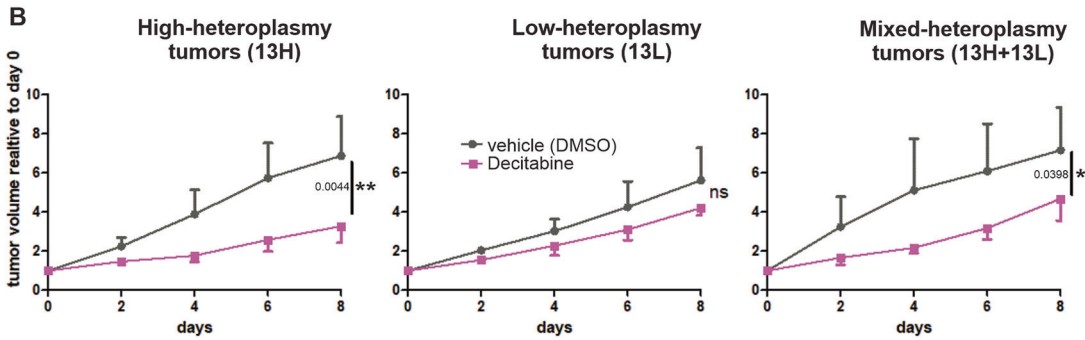

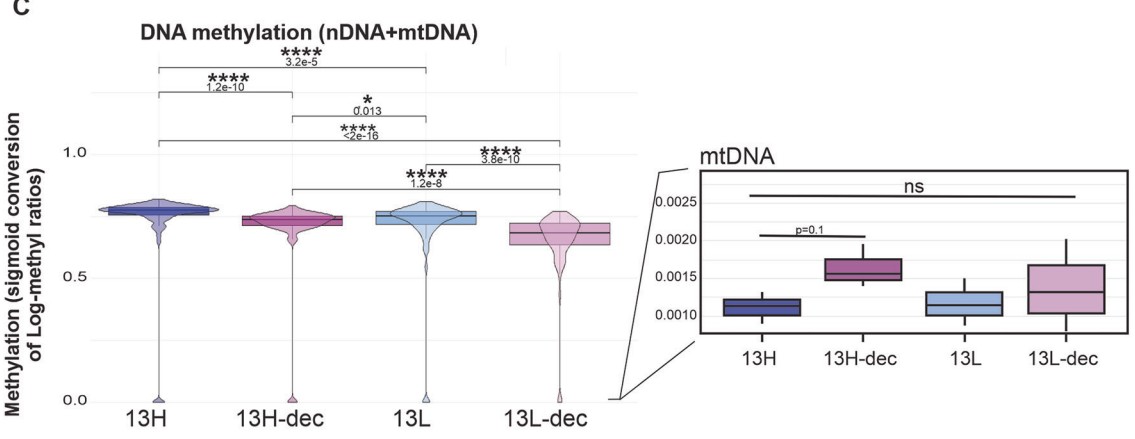

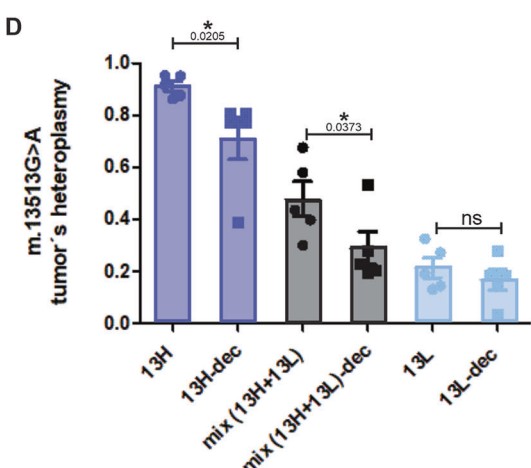

**Figure 8. Decitabine impacts tumor growth and m.13513 G > A heteroplasmy in a xenograft model.**

(A) Xenograft experiments sketch. (B) Tumor growth in the different tumor types and treatments. Decitabine decelerates the growth of the high and mixed heteroplasmy tumors. Tumors were measured every other day, and their growth was quantified relating to tumor volume at day 0. **$P < 0.01$, *$P < 0.05$, two-way ANOVA test. $N = 10$ mice (5 for decitabine group, 5 for control group). ns = not statistically significant. (C) Tumor DNA methylation assessment through Nanopore sequencing. To transform the log-methyl ratio (LMR) into a probability-like measure of methylation, we applied the logistic (sigmoid) function: $P = 1/(1+e^{-LMR})$. This transformation maps LMR values to a range between 0 and 1. Left panel shows values across all chromosomes (including mtDNA). All conditions show significant differences exhibiting, as in EPIC results, higher methylation in the 13H tumors and confirming decitabine's demethylating effect. Right panel highlights mtDNA's extremely low methylation values with no significant differences between conditions and no demethylating effect of decitabine, even showing a tendency of higher methylation in the treated tumors, ****$P < 0.0001$, *$P < 0.05$, Wilcoxon matched-pairs signed rank test of paired comparisons of the median methylation values for each chromosome. Horizontal lines denote the median, box limits indicate 25th and 75th percentiles and whiskers extend from the lowest/largest value. $N = 3$ tumors from each condition 13H, 13H-dec, 13L, 13L-dec. (D) m.13513 G > A heteroplasmy of the different tumors with and without decitabine treatment. Heteroplasmy was assessed using SYBR green-based RT-qPCR. *$P < 0.05$, one-tailed, unpaired Student's $t$ test. $N = 5$ tumors for each group. Bars represent mean $+/-$ SEM. Source data are available online for this figure.

### Apoptosis assays

Annexin V-FITC (BD Biosciences®, cat#BDB-556420) was used to measure apoptosis. Cells were taken up at the end of treatments and ~50,000 cells/condition were used. They were centrifuged, washed and then resuspended in 50 μl Annexin V buffer (10 mM HEPES, 140 mM NaCl, 2.5 mM CaCl2, pH 7.4) + 2 μl Annexin V-FITC. The samples were incubated in the dark at room temperature for 15 min and then 200 μl of ice-cold Annexin buffer was added (250 μl final volume). Fluorescence was evaluated through flow cytometry (FACSAria flow cytometer_BD®) using excitation 488 nm, emission 530/30 nm. Flow cytometry assays were analyzed using FlowJo v X.0.7® software. As a positive control, UV exposure during 30 min 4 h before the apoptosis assay was performed.

### Western blots

Cells were lysed with lysis buffer solution (TRITON 0.5%, NaCl 150 mM, EDTA 5 mM, Tris–HCl 1 M, pH 7.5 + Halt™ protease inhibitor cocktail) and proteins (20 μg, measured with Pierce™ BCA Protein Assay Kit, Thermofisher Scientific® cat#23227) were run on a 10% (DNMT1 and ESR1), 12% or gradient 4-20% (TGX Stain-free gels cat #456-8096, for OXPHOS rodent WB antibody cocktail) SDS–polyacrylamide gel and then transferred to a nitrocellulose membrane. The membranes were blocked in 5% lowfat milk-PBS solution and then incubated overnight at 4 °C with primary antibodies: anti-DNMT1:1000 (rabbit-antibodies.com®- cat#A307589), anti-ESR1 1:1500 (rabbit-CSB-PA11399A0Rb Cusabio®), total OXPHOS rodent WB antibody cocktail 1:1000 (mouse- ab110413- abcam®), anti-β-actin 1:5000 (mouse- Invitrogen® cat# MA1-140, clone 15G5A11/E2) anti-vinculin 1:5000 (mouse-, Sigma-Aldrich®, cat#CP74, clone V284) or 1:2000 (rabbit-Assay Genie®, cat#CAB2752) followed by PBS-Tween washes and secondary antibody incubation (1.5 h at room temperature): mouse (Gt anti-mouse IgG -H + L -Invitrogen®, cat#31430) 1:20,000 and rabbit (Gt anti-Rb IgG -H + L- Invitrogen®, cat#31460) 1:6000. Bands were developed using chemiluminescence (ECL BPSBioscience®), visualized with a LAS Fujifilm 4000 system (GE Healthcare Life Sciences®) and quantified using ImageJ® software.

### Oxygen consumption rate (OCR)

OCR was measured using a Seahorse XFp Extracellular Flux Analyzer (Agilent technologies®). Two days prior to the assay, cells were seeded at a density of 30,000 cells/well in wells B-G (wells A and H contained media only). Twenty-four hours afterward, DMSO or decitabine were added in triplicate conditions to a concentration of 1 μM. The XFp sensor cartridge was calibrated with calibration buffer overnight at 37 °C. The following day, cell culture medium was replaced with buffered Seahorse medium (Agilent cat#103680-100) supplemented with glucose, pyruvate, and glutamine to reach the same concentration as DMEM high glucose, and incubated for at least 1 h at 37 °C. Mito-stress test (Tan et al, 2015) was carried out, and measurements of endogenous respiration were measured following sequential addition of 5 μM Oligomycin (Sigma-Aldrich cat#75351), 1 μM Trifluoromethoxy carbonylcyanide phenylhydrazone (FCCP, Sigma-Aldrich cat# C2920), and 5 μM Rotenone (Sigma-Aldrich cat# R8875), plus Antimycin A (Sigma-Aldrich cat# A8674) 2.5 μM. Results were normalized to μg protein per well after the Seahorse run, protein was quantified using Lowry Method Bio-Rad® Detergent Compatible (DC) Protein Assay. Alternatively, results were normalized to cell number by Bio-Rad® TC20 Automated Cell Counter or DNA content.

### Intracellular ATP content

Cells were incubated with quinacrine dihydrochloride (Abcam® cat# ab145375) in cell culture medium at 10 μM concentration 30 min in the dark at 37 °C.

Afterward, they were taken up and passed through a cytometer, excitation: 488 nm; band pass filter 530/30 nm. A control for intracellular ATP content reduction was prepared using 100 μM CCCP + 5 μM Oligomycin (Sigma-Aldrich®) added concomitantly with quinacrine.

### Reactive oxygen species (ROS) measurement

2′,7′-Dichlorofluorescin diacetate (DCF-DA) (Sigma- Aldrich® cat# D6883) was used. The different conditions were loaded with the probe in cell medium at a 10 μM concentration and incubated at 37 °C for 30 min in the dark. Then, cells were taken up, rinsed with PBS and passed through flow cytometer (FACSAria flow cytometer_BD®), excitation: 488 nm; band pass filter 530/30 nm. A positive control using $H_2O_2$ (Sigma-Aldrich®) 3 mM during the incubation period with DCF-DA.

### Cell tracking

m.13513 G > A low-heteroplasmy cybrids (13L) were labeled with Green CMFDA (5-Chloromethylfluorescein Diacetate, abcam® cat#ab145459,) 10 μM at 37 °C for 45 min in the dark. Afterward, these dyed cells and undyed m.13513 G > A high-heteroplasmy cybrids (13H) were taken up and counted using Bio-Rad® TC20 Automated Cell Counter. Equal amounts were mixed and plated

before treatments. 24 h later, treatment with DMSO, 5-azacytidine and decitabine was initiated. After the 3-day treatment, cells were taken up and the % of FITC(+) cells was measured through flow cytometry (FACSAria flow cytometer_BD®), excitation: 488 nm; band pass filter 530/30 nm.

### Indirect immunofluorescence

Mitophagy evaluation assays: Cells were fixed with 4% Paraformaldehyde solution(Sigma-Aldrich cat#158127) 30 min at room temperature, following washes with PBS, and then quenched with $ClNH_4$ 50 mM (Biopack, cat# 2000167906) for 30 min at room temperature. Cells were then permeabilized with Albumin 2%/Saponin0.1% PBS solution (Biopack cat#2000946003), following which they were incubated with anti-MAP1 LC3B 1:200 (A7198, antibodies.com® rabbit) and mouse anti-human TOMM20 antibody 1:200 (LSBio®, clone AT1B2, cat#LS-C755581) overnight at 4 °C. The primary antibodies were rinsed and then membranes were incubated with secondary antibodies Goat Anti-Rabbit IgG H&L (Alexa Fluor® 488) cat# ab150077 1:750 and Goat Anti-Mouse IgG H&L (Alexa Fluor® 647) cat# ab150115 1:750, washed, and coverslips were mounted on glass slides using Mowiol (Sigma-Aldrich cat#475904-M) + Hoechst 34580 1:1000 (Thermo Scientific®, cat#62249) and examined by fluorescence confocal microscopy (Olympus Confocal Microscope FV1000-EVA®).

### Xenografts

Ten 6-week-old female NSG mice weighing ~25 g, were anesthetized with isoflurane 4% in $O_2$, and injected subcutaneously in their backs with m.13513 G > A cybrids (suspended in PBS) to generate three subcutaneous tumors in each mouse. $10^6$ high-heteroplasmy (13H), $10^6$ low heteroplasmy (13L) and $10^6$ mixed heteroplasmy (half high 13H and half low 13L) cybrids were implanted separately and distinctively in the back of each mouse. We decided to inject the three types of cells in each mouse to reduce inter-animal differences and to optimize the number of animals used for the experiment. When all tumors became palpable (13 days after injection) half the mice were selected for the control group and half for the treatment one. The experimental mice were treated with intraperitoneal decitabine 1 mg/kg dissolved in PBS every other day for 8 days (4 doses were applied). The controls were treated with the equivalent vehicle (DMSO in PBS). Mice were closely monitored, and tumor size was measured every other day. Tumor volume was calculated as= length*width²/2/1000. The mice were then euthanized in a $CO_2$ chamber, and tumors were excised. To obtain a reliable heteroplasmy value, representative to the whole tumor, tumors were homogenized in PBS with an Ultra turrax homogenizer and DNA was extracted from the resulting tissue pellet (sketch in Fig. 8A).

The highly immunosuppressed Nod Scid Gamma (NSG) mice (NOD.Cg-PrkdcscidIl2rgtm1Wjl/SzJ, NSG) (RRID:IMSR_JAX:005557) used in these experiments were obtained from Jackson Laboratory and were housed in a pathogen-free condition throughout the experimental duration. All procedures were performed following the consideration of animal welfare and were approved by the Institutional Committee for Care and Procedures of Laboratory Animals (CICUAL in Spanish) of the National University of Cuyo, Mendoza, Argentina. All procedures were approved by the Institutional Animal Care and Use Committee of the School of Medical Science, Universidad Nacional de Cuyo (Protocol approval N° 192/2021). All animals were cared for in accordance with the guiding principles in the care and use of animals of the US National Institute of Health.

## Statistics

GraphPad Prism v5.03® was used for most statistical analyses and graph confection.

EPIC studies required GenomeStudio Software® and ggplot2, tidyr, ggpubr R packages. Two-way ANOVA required nortest R package.

All experiments included at least three biological replicates. Statistical analyses were tailored to the design of each experiment, and the tests used are specified in the corresponding figure legends. All significant $P$ values are shown within the figures.

Regarding the animal experiments, specific randomization procedures were not applied; however, there were no significant differences in tumor size between control and treatment groups at the beginning of the treatment. Blinding was not implemented.

---

**The paper explained**

**Problem**

Mitochondrial diseases are genetic disorders that currently have limited treatment options, particularly when caused by mutations in mitochondrial DNA (mtDNA). Unlike nuclear genes, mtDNA exists in many copies per cell, and both mutated and non-mutated (wild-type) mtDNA molecules can coexist in the same cell—a condition known as *heteroplasmy*. Symptoms typically appear only when the proportion of mutated mtDNA exceeds a threshold. Therefore, some therapeutic strategies aim to reduce this ratio—a process known as "heteroplasmy shift"—to delay or prevent disease onset. However, current methods to directly remove or edit mutant mtDNA remain technically complex and difficult to apply clinically. In this study, we tested whether exploiting the natural communication between mitochondria and the nucleus—specifically nuclear DNA methylation—could offer a novel strategy for lowering heteroplasmy.

**Results**

We found that cells with high levels of mutant mtDNA develop a distinct nuclear DNA methylation profile that supports their survival. This pattern was less evident in cells with lower heteroplasmy. By treating cells with FDA-approved DNA methylation inhibitors—5-azacytidine and decitabine—we selectively impaired high-heteroplasmy cells, while sparing low-heteroplasmy cells. As a result, the overall proportion of mutant mtDNA decreased.

These effects were confirmed in both cultured cells and a xenograft mouse model, where treatment led to a significant reduction in heteroplasmy. The mechanism was traced to changes in nuclear—not mitochondrial—DNA methylation. These findings suggest that high-heteroplasmy cells rely on an epigenetic survival program, which can be disrupted to promote cells with healthier mitochondrial populations.

**Impact**

This study introduces a new therapeutic concept for mitochondrial disorders: reducing mtDNA heteroplasmy by targeting nuclear epigenetic mechanisms rather than attempting to edit the mitochondrial genome directly. Since the proposed drugs are already approved for clinical use, this strategy could be more readily translated to patients. While effective in proliferating cells and in vivo tumors, its impact in post-mitotic tissues remains uncertain. Further studies in physiological disease-relevant models are needed to evaluate therapeutic potential in affected organs. Beyond mitochondrial disorders, the selective vulnerability of cells with dysfunctional mitochondria to epigenetic modulation may also have implications for other conditions, such as cancer, where mitochondrial fitness influences cell survival.

## Graphics

Figures were created with CorelDRAW 2021®.

MS-MLPA heatmap (Fig. 2A) was confectioned using Complex Heatmap R package.

Violin plots and box plots in Fig. 3B were prepared with ggplot2, tidyr y ggpubr R packages.

EPIC heatmap (Fig. 3C) was prepared with heatmapper.ca (Babicki et al, 2016).

Figure 8 contains images modified from Biorender®.

## Article drafting

We used ChatGPT, an AI-powered language model developed by OpenAI, solely for language-related assistance in the composition of this research paper, with no influence on the content or research outcomes.

# Data availability

All data supporting the findings of this study are available in the main text, supplementary material and GitHub repository. The datasets produced in this study are available in the following databases: EPIC 850k Methylation data: Gene Expression Omnibus (GEO) GSE300902. Nanopore methylation data: https://www.mediafire.com/file/q9xjh6it3t0cu7s/Tabix.tsv.bgz/file. R scripts: GitHub repository: https://github.com/slaurito/Methylation-Analysis-from-Nanopore-and-Epic-Data-Perez-et-al. Any additional information can become available upon request.

The source data of this paper are collected in the following database record: biostudies:S-SCDT-10_1038-S44321-025-00285-5.

# Peer review information

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

## Acknowledgements

First, we thank the United Mitochondrial Disease Foundation, especially the families that voted for this project, and of course, the donors that contributed with the funding for the accelerator prize. This prize made the project possible. We acknowledge the rest of the funding that also contributed to this work. We also thank Dr. LS Mayorga (statistical consulting, manuscript revision—IHEM, Mendoza, Argentina) and Dr M Roqué (manuscript revision—IHEM, Mendoza, Argentina). We thank the core facilities that supported this project: the Instituto de Histología y Embriología de Mendoza (IHEM, CONICET, UNCuyo) and the Department of Neurology at the University of Miami Miller School of Medicine. We acknowledge our funding resources: 2021 UMDF Accelerator prize: Modulation of the nuclear epigenome as a new strategy for mtDNA heteroplasmy shift. Lía Mayorga. Wood-Whelan research fellowship 2019 for a short stay at Moraes Lab (Miller School of Medicine, University of Miami, USA). "Modulation of the epigenome as a new strategy for mtDNA heteroplasmy shift". PICT 2019-00449 (PICT joven 2019): Shift de la heteroplasmia mitocondrial por modulación del epigenoma nuclear: posible tratamiento para enfermedades mitocondriales. Programa de Redes Federales de Alto Impacto, proyecto de investigación: Genómica Clínica de Enfermedades Poco frecuente, Ministerio de Ciencia, Tecnología e Innovación de la Nación Argentina. 2023. The work in the Moraes Lab is funded by the National Institutes of Health (NIH) award R01EY010804, the Army Research Office (W911NF-21-1-0248), the Muscular Dystrophy Association (MDA 964119), The Research to Prevent Blindness Stein award and the Florida Biomedical Foundation (21K05).

## Author contributions

**María J Pérez**: Formal analysis; Investigation; Methodology; Project administration; Writing—review and editing. **Rocío B Colombo**: Formal analysis; Investigation; Methodology; Writing—review and editing. **Sebastián M Real**: Supervision; Investigation; Methodology; Writing—review and editing. **María T Branham**: Conceptualization; Data curation; Formal analysis; Investigation; Writing—review and editing. **Sergio R Laurito**: Data curation; Formal analysis; Investigation; Methodology; Writing—review and editing. **Carlos T Moraes**: Formal analysis; Supervision; Funding acquisition; Writing—review and editing. **Lía Mayorga**: Conceptualization; Data curation; Formal analysis; Funding acquisition; Validation; Investigation; Methodology; Writing—original draft; Project administration; Writing—review and editing.

Source data underlying figure panels in this paper may have individual authorship assigned. Where available, figure panel/source data authorship is listed in the following database record: biostudies:S-SCDT-10_1038-S44321-025-00285-5.

## Disclosure and competing interests statement

The authors declare no competing interests.

# Expanded View Figures

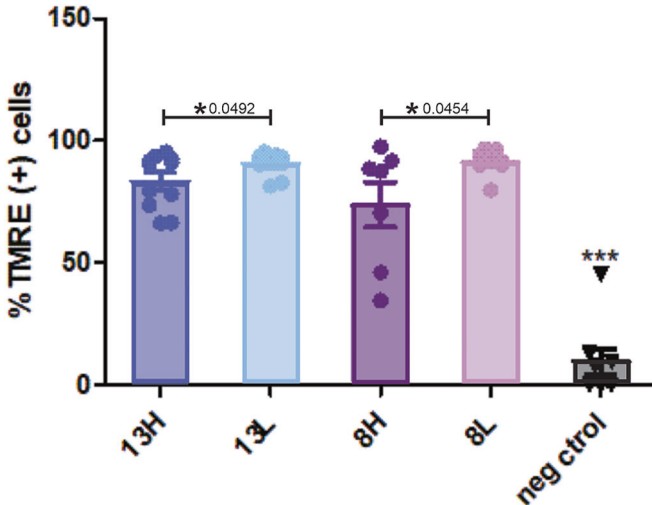

**Figure EV1. Differences in mitochondrial membrane potential between high and low heteroplasmy cybrids.**

Mitochondrial membrane potential measured with TMRE through Flow cytometry. Using a threshold based on autofluorescence in the TMRE-PE histogram, cells were designated as TMRE (+) or (−) for quantification. One-tailed unpaired Student´s *t* tests. N: 13H = 11, 13 L = 9, 8H = 7, 8 L = 7 independent experiments. *$P < 0.05$. Negative control CCCP. ***$P < 0.001$. Negative control significance is shown compared to the condition with the less difference (8H). Bars represent mean $+/-$ SEM.

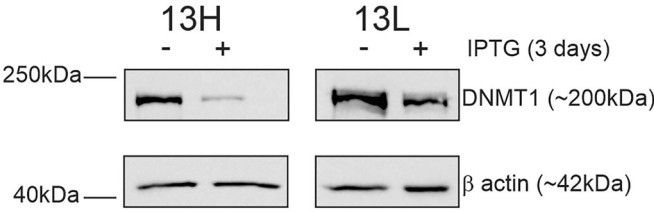

**Figure EV2. DNMT1 knockdown control in ´13´ cybrids.**

DNMT1 expression through Western Blot in the stable cell line of m.13513 G > A low (13 L) and high-heteroplasmy (13H) cybrids with the mammalian pLV[shRNA]-LacI: T2A:Puro-U6/2xLacO > hDNMT1[shRNA#1]. 3-day treatment with IPTG: isopropyl-galactosidase was installed to decrease DNMT1 expression.

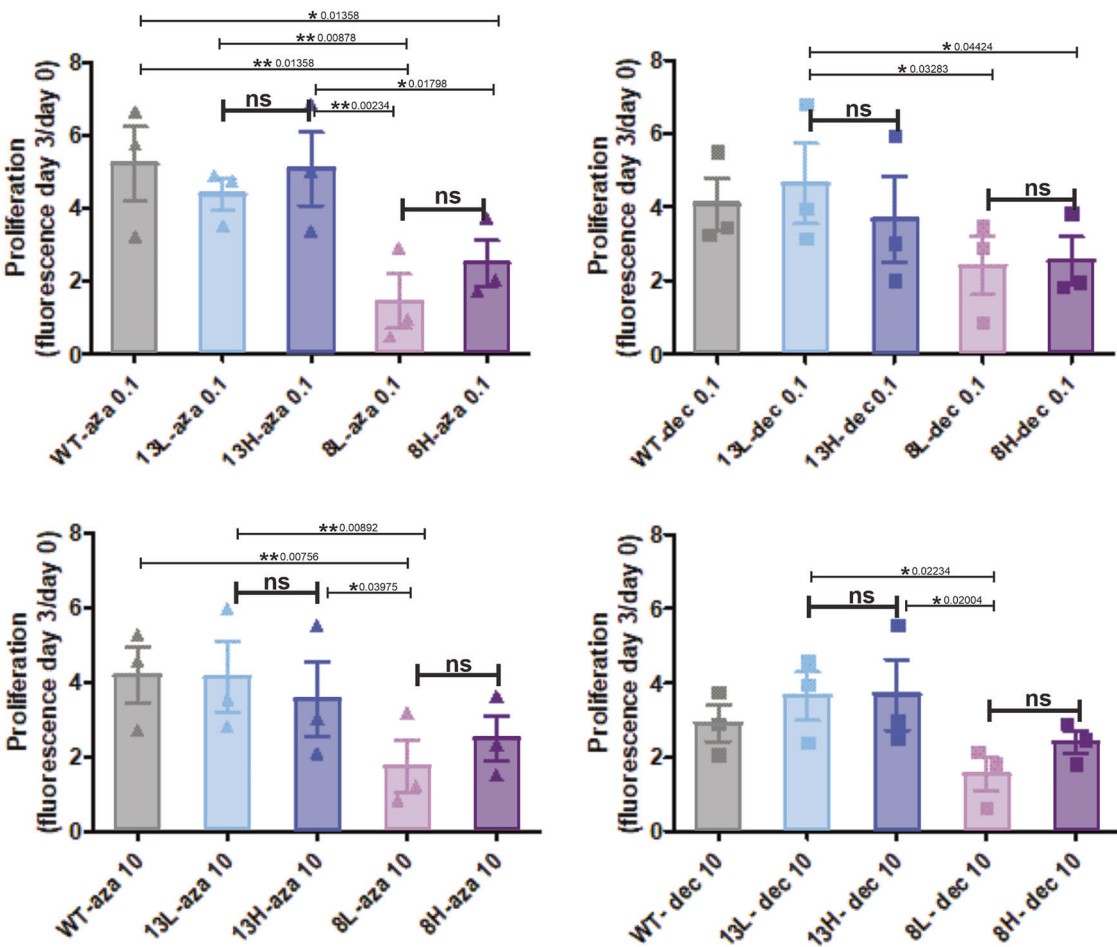

**Figure EV3.** **DNA methylation inhibitors reduce differences in proliferation rates of high-low-heteroplasmy cybrids.**

Differential proliferation of cells treated with 5-azacytidine (aza) and decitabine (dec) 0.1 μM (upper panels) or 10 μM (lower panels) for 3 days. Two-way ANOVA with fixed categorical variables, followed by Tukey's Honest Significant Difference test for post hoc comparisons. Residual normality was assessed using the Shapiro–Wilk test, implemented with the tidyr and nortest packages in R. All conditions passed the normality test. Comparisons between high and low heteroplasmies are stood out to compare with Fig. 4A. ns= not significant. **$P < 0.01$, *$P < 0.05$. $N = 3$ independent experiments, with technical duplicates. Bars represent mean $+/-$ SEM.

**A**

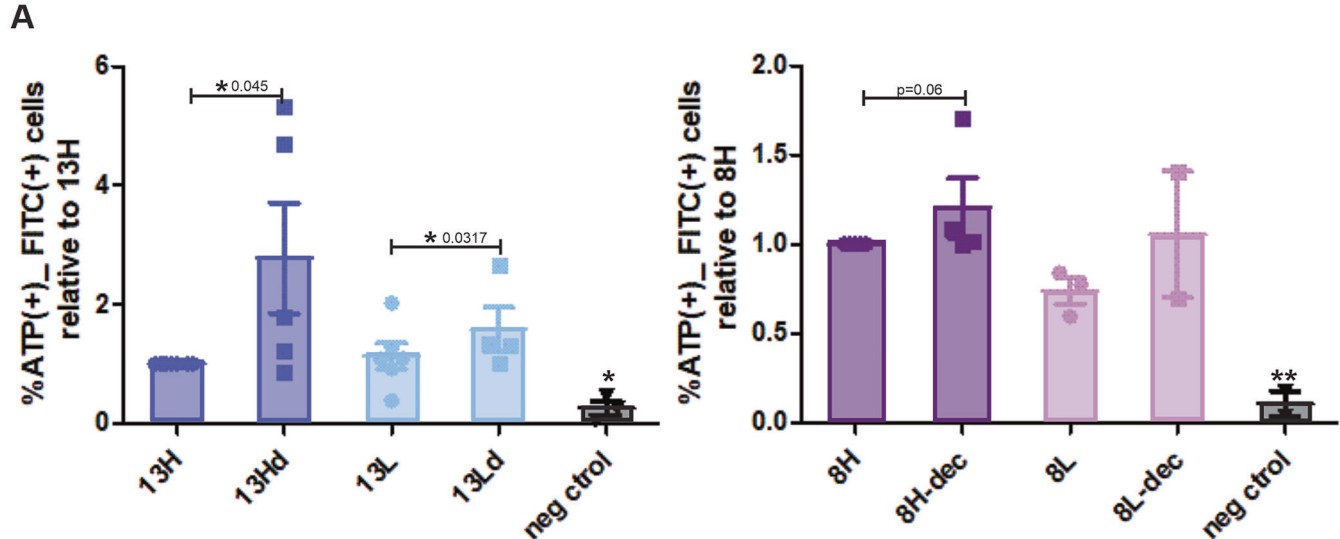

**B**

**C**

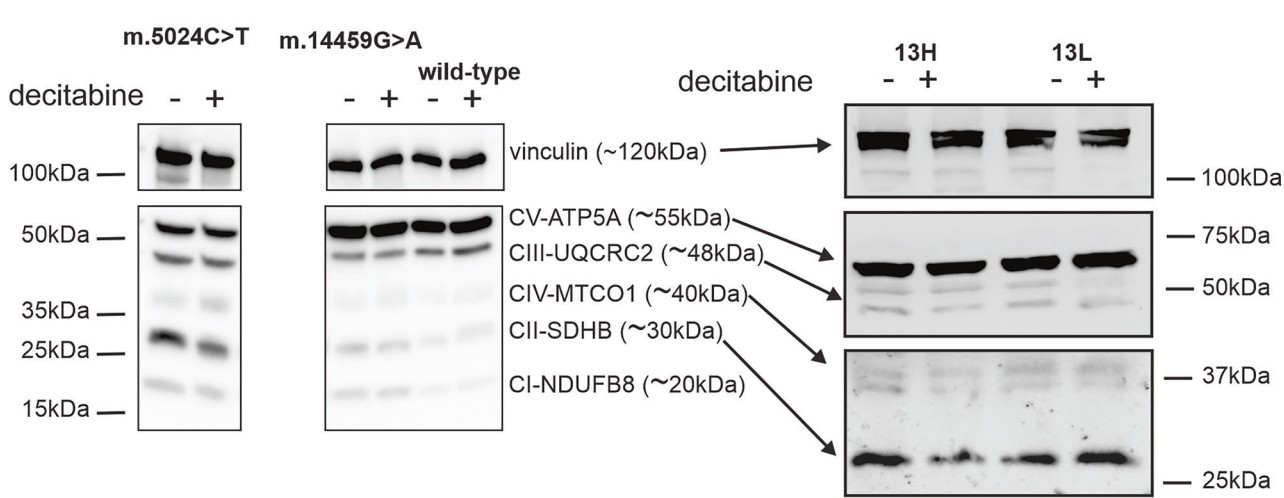

◀

**Figure EV4. DNA methylation inhibitor decitabine modifies mitochondrial function without altering the expression of mitochondrial complex proteins.**

(A) Intracellular ATP content measured with Quinacrine through flow cytometry in cybrids treated 3 days with 1 μM. A threshold based on autofluorescence was set to divide cell populations as ATP (+) or (−) = FITC (+) or (−). The graph was outlined using values relative to the high-heteroplasmy sample (13H or 8H). *$P < 0.05$. One-tailed Wilcoxon matched-pairs signed rank test between unnormalized values of intra-experimental conditions. For 13 cybrids 3–7 independent experiments, for 8 cybrids 2–5. ATP content is shown significantly increased after treatment with decitabine in the 13 cybrids. A tendency to the same is shown in the 8. A control for ATP reduction was performed using CCCP+ Oligomycin, its′ significant fluorescent reduction is shown comparing to the sample with the lowest mean FITC (+) (13H and 8 L respectively), *$P < 0.05$, **$P < 0.01$. (B) Mitochondrial membrane potential measured with TMRE through Flow cytometry. Using a threshold based on autofluorescence in the TMRE-PE histogram, cells were designated as TMRE (+) or (−) for quantification. $N = 4$–11 independent experiments. One-tailed unpaired Student′s $t$ tests. *$P < 0.05$, **$P < 0.01$. Negative control CCCP. ***$P < 0.001$. Negative control significance is shown compared to the condition with the less difference (8H). Bars represent mean +/− SEM. (C) Western blot to study the expression of mitochondrial complexes in different cells with and without decitabine treatment 1-day 1 μM. Left panel shows a gradient 4-20% SDS–polyacrylamide gel, right panel 12% SDS–polyacrylamide gel. In the 12% gel, CI-NDUFB8 was not visible. m.5024 C > T ~ 50% heteroplasmy MEFs, m.14459 G > A fibroblasts~95% heteroplasmy. 13H= m.13513 G > A high-heteroplasmy~80%, 13 L m.13513 G > A low-heteroplasmy~20%.

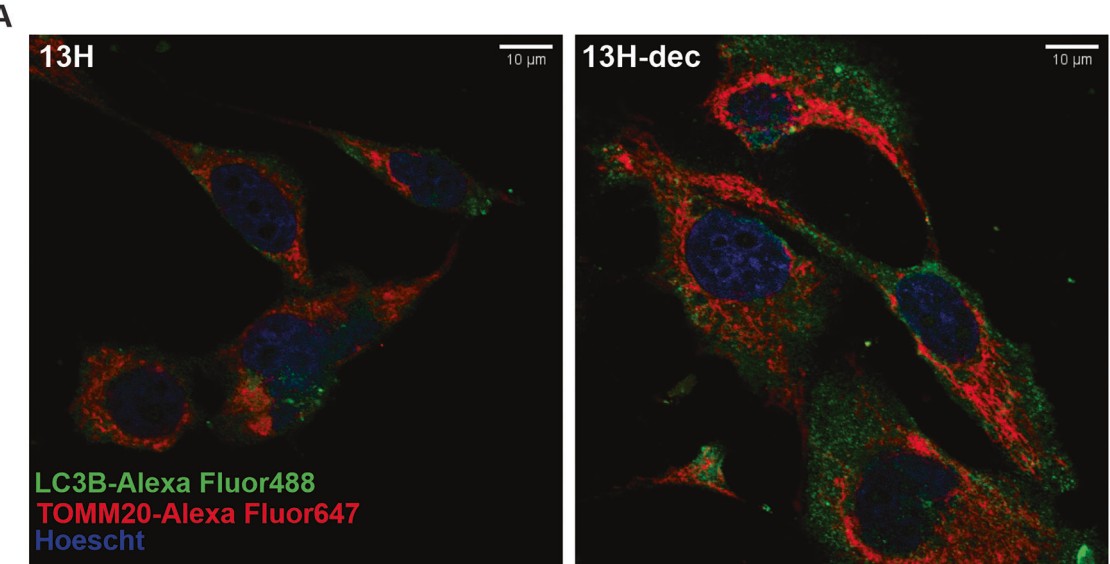

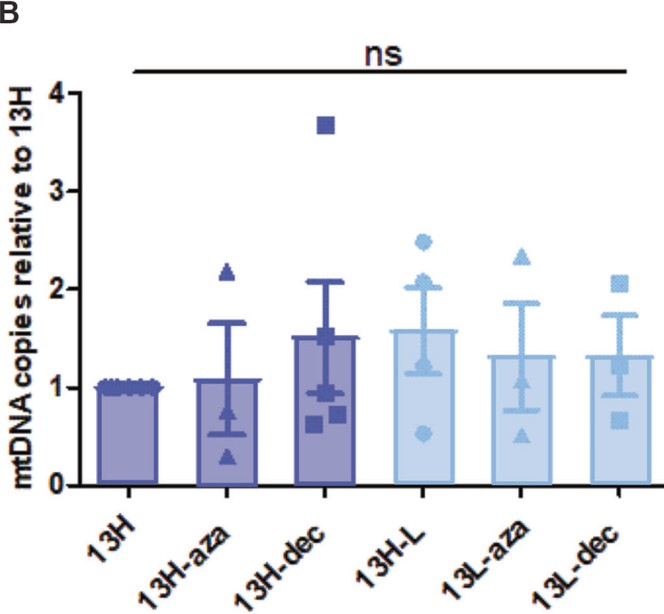

**Figure EV5.  DNA methylation modulators do not impact mitophagy nor mtDNA content.**

(A) Confocal Immunofluorescence Microscopy. Confocal laser micrographs depict indirect immunofluorescence staining of LC3B (labeled with Alexa Fluor 488 in green) and TOMM20 (labeled with Alexa Fluor 647 in red) within m.13513 high-heteroplasmy cybrids (13H) treated with decitabine 1 µM—3 days (right panel) or DMSO (left panel). The cells´ nuclei were stained with Hoescht (blue). Colocalization of the two proteins is not evidenced. (B) mtDNA content. mtDNA content was measured with SYBR green-based RT-qPCR relativizing the m.13513 ´all´ amplicon to the nuclear gene B2M, quantified using ΔCT method= $2^{-(allCT-B2MCT)}$. Kruskal–Wallis test, Dunn's Multiple Comparison Test. $N = 3$–5 experiments. Bars represent mean $+/-$ SEM.

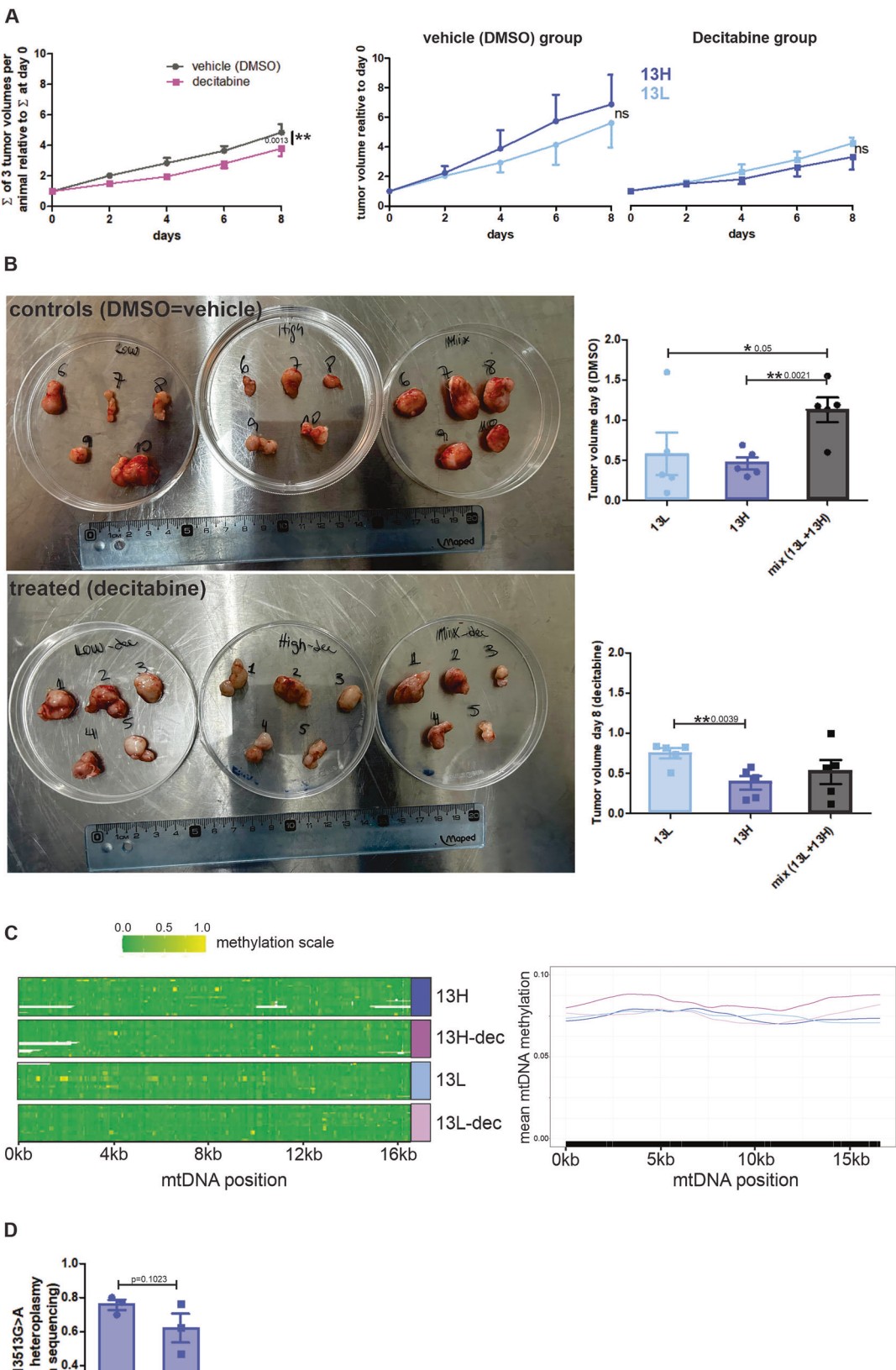

◀ **Figure EV6.  Further characterization of xenograft experiments regarding tumor growth, tumor volume, mtDNA methylation and heteroplasmy.**

(A) Left panel: overall tumor growth in mice in the control (DMSO) and treated (Decitabine) group. The volumes of the 3 tumors per animal were added up every measurement day and that sum was relativized to the sum of tumor volume of day 0 per animal. $**P < 0.01$, two-way ANOVA test. Right panel: comparison of the high and low-heteroplasmy tumor growth within the control and treated group. (B) Final tumor size assessment at day 8. It is worth clarifying that tumor sizes started out uneven in order to have all 10 mice under treatment at the same time. Therefore, final tumor volumes *do not* represent their growth rate. Anyhow, we can stand out that the mixture of 13H + 13L grew to higher volumes (many ulcerating the skin -3 out of 5 mice-) when left untreated, upper panel. Also, within the treated mice (lower panel), the low-heteroplasmy tumors were less affected by decitabine than the high-heteroplasmy ones, $*P < 0.05$. One-tailed, unpaired Student´s *t* test. (C) mtDNA methylation assessment using Nanopore sequencing technology. Coverage of the whole mitochondrial genome was satisfactory and showed low methylation values that were similar in all four conditions. No significant differences were observed, however, paradoxically the 13H-dec condition showed higher trend in methylation values than its untreated counterpart. $N = 3$ tumors from each condition: 13H, 13H-dec, 13L, 13L.dec. (D) Heteroplasmy assessment through Nanopore sequencing. Base call counts at position m.13513 are available in Dataset EV2. Heteroplasmy was calculated as ´A´ counts/(´A´counts + ´G´ counts). $N = 3$ tumors from each condition: 13H, 13H-dec, 13L, 13L.dec. Bars represent mean $+/-$ SEM.

