## [Peer Review File · EMBO Molecular Medicine]

Rewriting nuclear epigenetic scripts in mitochondrial diseases, a strategy for heteroplasmy control

María Pérez, Rocío Colombo, Sebastián Real, María Branham, Sergio Laurito, Carlos Moraes, and Lía Mayorga

Corresponding author: Lía Mayorga (liamayorga@fcm.uncu.edu.ar)

Review Timeline:

Submission Date:	1st Apr 25
Editorial Decision:	2nd May 25
Revision Received:	18th Jun 25
Editorial Decision:	2nd Jul 25
Revision Received:	14th Jul 25
Accepted:	17th Jul 25

Editor: *Zeljko Durdevic*

Transaction Report:

2nd May 2025

Dear Dr. Mayorga,

Thank you for the submission of your manuscript to EMBO Molecular Medicine. We have now received feedback from the three reviewers who agreed to evaluate your manuscript. All three referees recognize interest of the study but also raise serious and partially overlapping concerns that should be addressed in a major revision. If you would like to discuss further the points raised by the referees, I am available to do so via email or video. Let me know if you are interested in this option.

We would welcome the submission of a revised version within three months for further consideration. Please let us know if you require longer to complete the revision.

I look forward to receiving your revised manuscript.

Yours sincerely,

Zeljko Durdevic

We require:

- 1) A .docx formatted version of the manuscript text (including legends for main figures, EV figures and tables). Please make sure that the changes are highlighted to be clearly visible.
- 2) Individual production quality figure files as .eps, .tif, .jpg (one file per figure). For guidance, download the 'Figure Guide PDF': (<https://www.embopress.org/page/journal/17574684/authorguide#figureformat>).
- 3) A .docx formatted letter INCLUDING the reviewers' reports and your detailed point-by-point responses to their comments. As part of the EMBO Press transparent editorial process, the point-by-point response is part of the Review Process File (RPF), which will be published alongside your paper.
- 4) A complete author checklist, which you can download from our author guidelines (<https://www.embopress.org/page/journal/17574684/authorguide#submissionofrevisions>). Please insert information in the checklist that is also reflected in the manuscript. The completed author checklist will also be part of the RPF.
- 5) Please note that all corresponding authors are required to supply an ORCID ID for their name upon submission of a revised manuscript.
- 6) It is mandatory to include a 'Data Availability' section after the Materials and Methods. Before submitting your revision, primary datasets produced in this study need to be deposited in an appropriate public database, and the accession numbers and

database listed under 'Data Availability'. Please remember to provide a reviewer password if the datasets are not yet public (see <https://www.embopress.org/page/journal/17574684/authorguide#dataavailability>).

12) Author contributions: You will be asked to provide CRediT (Contributor Role Taxonomy) terms in the submission system. These replace a narrative author contribution section in the manuscript.

13) A Conflict of Interest statement should be provided in the main text.

14) Every published paper now includes a 'Synopsis' to further enhance discoverability. Synopses are displayed on the journal webpage and are freely accessible to all readers. They include a short stand first (maximum of 300 characters, including space) as well as 2-5 one-sentences bullet points that summarizes the paper. Please write the bullet points to summarize the key NEW findings. They should be designed to be complementary to the abstract - i.e. not repeat the same text. We encourage inclusion of key acronyms and quantitative information (maximum of 30 words / bullet point). Please use the passive voice. Please attach these in a separate file or send them by email, we will incorporate them accordingly.

15) Include a Reagents and Tools Table as part of the Methods section, which can be downloaded from our author guidelines (<https://www.embopress.org/page/journal/17574684/authorguide#structuredmethods>)

***** Reviewer's comments *****

Referee #1 (Comments on Novelty/Model System for Author):

The model systems were transmitochondrial cybrids and xenografts. The results are very interesting but unsure how this can be extrapolated to post-mitotic tissues which are often the most affected tissue in mitochondrial disease.

Referee #1 (Remarks for Author):

This manuscript from Perez et al. reports a series of intriguing expts to show that inhibition of nuclear genome methylation causes a rescue of mtDNA heteroplasmy both in cell lines and xenografts. By using different mtDNA mutant cell lines and levels of heteroplasmy, those with high levels of heteroplasmy show an increase in nuclear genome methylation. On treatment of these high het lines with methylation inhibitors, high het cells appear to be lost due to increased apoptosis, resulting in a reciprocal increase in low het cells. The data is largely impressive and supportive of this hypothesis. The authors also include an important section on limitations of this technique. Perhaps the most important caveat is that whilst this is undoubtedly an impressive and surprising observation, it is only applicable to dividing cells and much of the effect of mtDNA disease is on post-mitotic tissue such as muscle and CNS. However, I think this finding is very important and the data supports the hypothesis. I have a few questions the authors may wish to address:

1. In Fig4A, why are the WT-Aza and 13L-Aza treated fluorescent signals so high? Although statistical analysis is used, it still raises some concern over the conclusions drawn from this expt.

2, I would recommend adding the oxygen consumption data from the other cell lines to Fig4. The authors do include them in a supplemental figure and the text explains that all lines showed the oxygen consumption increase, but I think its important to add them as the assumption made merely by looking at this figure as it stands is that only the high-het line has oxygen consumption increased by methylation inhibitor treatment.

3. Why does methylation inhibition lead to an increase in mitochondrial ROS in all instances? I don't expect the authors to know this, but its an important observation.

Referee #2 (Remarks for Author):

The corresponding author of this manuscript had previously reported that mitochondrial stress induces epigenetic modifications of nuclear DNA (ref 23). Here, the authors propose that blocking the adaptative nDNA methylation pattern that is triggered by mitochondrial dysfunction (which favours survival of cells with high mtDNA mutation load), will result in impaired viability of these cells, thus favouring cells with low levels of mtDNA mutation load. Using cultured cybrids with pathogenic mtDNA mutations at different mutation loads the authors investigate the association between the methylation pattern observed in nDNA and the degree of heteroplasmy. After identifying several differentially methylated genes, somehow involved in cell survival, the authors experimentally test how inhibition of DNA methylation (using 5-azacytidine and decitabine) results in reduced survival of cybrids with high mutation load. They finally propose that this strategy could promote a heteroplasmic shift towards lower heteroplasmic load of pathogenic mutations in patients.

While I think that most experiments are convincing and the overall results support their conclusions (which are relevant in the area of mitochondrial genetics), some inaccuracies in the presentation in the results pose some confusion in the interpretation of their observations and in the transparency of the presentation of the results. The main issue relates to the statistical analyses of some experiments. Often (some cases specifically pointed in my comments below) the authors use parametric methods to compare groups with very few replicates. Since the application of parametric methods have specific requirements regarding the

distribution of the results analysed (normality and homogeneity of variances, in most cases), its use should be avoided for experiments with so low number of replicates, for which it cannot be determined if data meets these requirements. Nonparametric methods should be used instead. It is true that nonparametric tests are less powerful than parametric tests to detect differences, but the authors can still discuss about their data and the observed values, independently of whether the number of experimental data is sufficient to reach statistical significance or not. In addition, there are several comparisons in which the authors used paired tests for experiments in which the individual results of one group do not seem to be paired with the individual results of the compared group or groups.

Also, I could not find anywhere indications about what the error bars state for in the plots. Either the authors should indicate in the methodology section what error bars represent (if it is the same for all plots) or they should explicitly indicate it in each figure legend.

In most cases, the individual results are shown as dots in the plots (with few exceptions, some of them pointed below) and the readers can see the extent of the differences. Forcing the statistics to show significance with use of tests for results that do not meet their requirements is not accurate.

Some specific examples of these points:

-Page 11 (Fig 2): We need to know what the error bars stand for. Using Student t test for $N \geq 2$ is too forced. A nonparametric test should be used. In addition, for so low numbers, bar and error bars is not informative enough. It would be best plotting the actual values as dots (as done in Figs 4, 5, 6 etc). Another question is: why they use paired analyses? Unless there is an experimental link between each one-to-one specific result paired analyses cannot be used.

-Page 15, Fig 4A: Again, it is not clear why the test used to compare different cell lines was a repeated measures ANOVA. The groups that are compared in this figure do not derive from repeated or related measures. The only repeated measure I can see here is fluorescence at days 0 and 3, and these two repeated measures are used to calculate the ratio (each data value in the plot). The same applies to Fig S3. Fig S3 should show the individual results, as Fig. 4 does. In addition, I do not think comparing groups of $N=3$ through a parametric method such as ANOVA is accurate because it cannot be ensured that a sample of 3 results meets the requirements for the use of parametric methods (i.e., derive from a normal distribution and with homogeneous variances among groups). A nonparametric test would be more appropriate. As mentioned above, nonparametric tests are less powerful than parametric tests to detect differences, but the authors can still discuss about their data and the observed values, independently of whether the number of experimental data is sufficient to reach statistical significance or not. For example, for the DMSO group, perhaps nonparametric methods cannot detect significant differences between 13H and 13L (or between 8H and 8L), but the readers will be aware of the data and how the discussion and conclusions are supported or not by the data.

-Page 18, Fig 6B. The authors use a parametric test for comparing groups with only $N=3$. Also related to the 6B plots: why a paired statistical test was used? If there is an experimental link between each DMSO result with a corresponding -dec result (for example, 3 pairs of different one-to-one comparisons done in three different days), then it would be more informative and transparent for readers to link each related pair with a line. If they are independent measurements, then a paired comparison should not be used. The same applies to panel 6C.

-Page 20, Figure 7A: Again, a paired test is used for an experiment in which each result does not seem to be linked to another specific result of the compared group. When results are linked, the link should be identified for transparency. If the authors consider each individual experiment pairs each 13H-untreated result to a specific 13H-dec result, as one pair of linked results ($N=5$), why the central column only has $N=3$? The same applies to 7D (6-8 independent experiments).

Other comments:

-Page 11 (Fig 2): Methylation levels of 8L and 8H are not different and much lower than WT. How the authors explain this?

-The cybrids were always cultured in a medium with high content in glucose. In these conditions, the cybrids entirely rely on glycolysis and pyruvate reduction to lactate, with no functional dependence on the oxidative phosphorylation. Did the authors try to replicate some experiments using galactose instead of glucose? This would make the cells more dependent on OXPHOS, which may have consequences in terms of tolerance and survival of high heteroplasmy of mutations.

Minor points:

-Page 2: the authors enumerate: "succinate, fumarate, NAD, NADH and reactive oxygen species". In this context, when they say "NAD", I think they refer to the cationic form NAD^+ , because the acronym "NAD", without the plus sign, means "nicotine amide dinucleotide", which includes both NAD^+ and NADH. Therefore, I think it should be written " NAD^+ ".

-Page 15 (Figure 4A): The authors may consider switching the hierarchy of the X axis label. Since the measured variable of interest is cell proliferation, and the "units" used to quantify cell proliferation is the ratio "fluorescence at day 3/day0" (which are the actual values indicated in the axis numbers), perhaps it would make more sense to label "Proliferation (fluorescence at day 3/day0)". The same applies to Fig. S3.

Referee #3 (Comments on Novelty/Model System for Author):

This manuscript presents a comprehensive and insightful analysis of the epigenetic alterations associated with mitochondrial DNA (mtDNA) mutations. The authors also provide compelling evidence that interventions aimed at reversing nuclear DNA methylation may lead to a reduction in heteroplasmy levels, both in cellular systems and in vivo models. The experiments investigating DNA methylation are of particularly high quality and contribute meaningfully to the field.

However, some aspects of the study would benefit from further refinement. Specifically, the analyses related to mitochondrial respiration, reactive oxygen species (ROS) production, and respiratory chain protein expression appear to lack appropriate controls, which makes it challenging to fully assess the significance and robustness of these findings.

A major concern pertains to the translational relevance of the proposed therapeutic strategy. As tissues primarily affected in mitochondrial diseases are largely post-mitotic, it remains unclear how inducing apoptosis in high-heteroplasmy cells would confer a therapeutic benefit in such contexts. To strengthen the translational impact of the work, the inclusion of additional experiments using more physiologically relevant or disease-representative models is strongly recommended.

Referee #3 (Remarks for Author):

In this study, Perez and colleagues investigate the potential of epigenetic modulators to alter heteroplasmy levels in cybrid models harboring various mtDNA mutations. The authors initially demonstrate that cells with a high proportion of mutant mtDNA acquire a distinctive nuclear DNA methylation profile that is critical for their survival and not observed in cells with low heteroplasmy. Utilizing FDA-approved inhibitors of DNA methylation, the authors selectively compromised the viability of high-heteroplasmy cells, resulting in an enrichment of cells with lower mutant loads and an overall reduction in heteroplasmy. These findings were validated in both cellular and animal models, highlighting a previously underappreciated role of nuclear DNA methylation in supporting the persistence of cells experiencing severe mitochondrial dysfunction. This work suggests that targeting nuclear epigenetic mechanisms may represent a promising therapeutic avenue for mitochondrial diseases, potentially expanding the current landscape of treatment strategies beyond direct mtDNA manipulation.

While the study is innovative and presents a compelling concept, I have several points of concern, particularly regarding the translational applicability of the proposed intervention. Since mitochondrial diseases primarily affect post-mitotic tissues, it remains unclear how selectively inducing apoptosis in high-heteroplasmy cells would provide therapeutic benefit in such non-proliferative contexts. Additional investigations using more physiologically relevant models could help address this important question.

Furthermore, given that the authors' strategy appears to hinge on promoting apoptosis to reduce heteroplasmy, it would be informative to explore whether high-heteroplasmy cells exhibit higher sensitivity to apoptotic stimuli more generally. Experiments utilizing alternative apoptosis inducers could help clarify this point. Along the same lines, it would be of considerable interest to assess the effect of blocking apoptosis-e.g., with Z-VAD-FMK or similar inhibitors-on the efficacy of 5-azacytidine or decitabine in lowering heteroplasmy.

I also have some concerns regarding the analysis of mitochondrial function. In Figures 6 and S4, the authors report that decitabine increases oxygen consumption in wild-type (WT) cybrids, MEFs, and 13H cells. It would enhance interpretability if data from WT and 13H cybrids were plotted on the same graph to facilitate direct comparison. Moreover, clarification is needed on what is meant by "WT cybrids"-does this refer to WT 143B cells, or to cybrids with 0% heteroplasmy? The reported increase in respiration and ATP production appears somewhat inconsistent with the concurrent rise in ROS, as improved mitochondrial coupling would typically be associated with decreased ROS generation. It would be valuable to know whether the authors measured mitochondrial membrane potential to support their interpretations.

Lastly, the analysis of respiratory chain proteins by western blot could be further strengthened by including the corresponding wild-type (or low-heteroplasmy) controls for MEFs and 13H cells, thereby providing a more complete and comparative perspective.

Minor comments:

In the Introduction, the statement "We hypothesized that manipulating the nDNA methylome would negatively impact the high-heteroplasmy cells, favor the prevalence of low heteroplasmy ones and consequently, reduce heteroplasmy" is somewhat

unclear. A rephrasing for clarity is recommended.

The abbreviations 13H/L and 8H/L used throughout the manuscript are a bit confusing and could be more clearly defined or replaced with more intuitive nomenclature to aid the reader.

RESPONSE TO REVIEWERS

We thank all three reviewers for their thorough and constructive comments. The feedback helped us clarify and strengthen the manuscript. Below we provide a detailed, point-by-point response to each. Reviewer points are shown in blue, and our responses are in black.

**** Reviewer's comments ****

Referee #1 (Comments on Novelty/Model System for Author):

The model systems were transmitochondrial cybrids and xenografts. The results are very interesting but unsure how this can be extrapolated to post-mitotic tissues which are often the most affected tissue in mitochondrial disease.

We thank the reviewer for raising this important point. We agree that the use of dividing cell systems represents a limitation when considering the translational potential of our findings, particularly given the vulnerability of post-mitotic tissues in mitochondrial disease. This limitation is addressed in the discussion section. We hypothesize that combining nuclear DNA demethylation approaches with strategies aimed at regenerating post-mitotic tissues could enhance therapeutic efficacy in these clinically relevant scenarios. In any case, we believe our findings provide proof-of-principle, that at least in some cells, epigenetic changes in the nucleus can modulate mtDNA heteroplasmy.

Referee #1 (Remarks for Author):

This manuscript from Perez et al. reports a series of intriguing expts to show that inhibition of nuclear genome methylation causes a rescue of mtDNA heteroplasmy both in cell lines and xenografts. By using different mtDNA mutant cell lines and levels of heteroplasmy, those with high levels of heteroplasmy show an increase in nuclear genome methylation. On treatment of these high het lines with methylation inhibitors, high het cells appear to be lost due to increased apoptosis, resulting in a reciprocal increase in low het cells. The data is largely impressive and supportive of this hypothesis. The authors also include an important section on limitations of this technique. Perhaps the most important caveat is that whilst this is undoubtedly an impressive and surprising observation, it is only applicable to dividing cells and much of the effect of mtDNA disease is on post-mitotic tissue such as muscle and CNS. However, I think this finding is very important and the data supports the hypothesis. I have a few questions the authors may wish to address:

1. In Fig4A, why are the WT-Aza and 13L-Aza treated fluorescent signals so high?

While we do not have a definitive explanation for this, there is evidence that 5-azacytidine, although primarily used as an anti-tumor agent^{1,2}, can transiently enhance proliferation by upregulating stemness-associated transcription factors³. Furthermore, in our system, 13L cybrids did not show increased apoptosis in response to 5-

azacytidine, suggesting relative resistance. Although WT cybrids were not tested for apoptosis sensitivity, they may exhibit a similar response to 13L cells. Together, these factors could account for the unexpectedly high proliferation observed in the WT-aza and 13L-aza conditions.

Although statistical analysis is used, it still raises some concern over the conclusions drawn from this expt.

This point was also raised by referee #2. We have three independent experiments of this assay, each with technical duplicates. Mean values of duplicates from each independent experiment were plotted.

Following the reviewers' suggestions, we consulted a statistics expert and revised our approach by applying a two-way ANOVA with fixed categorical variables, followed by Tukey's Honest Significant Difference test for post-hoc comparisons. Residual normality was assessed using the Shapiro-Wilk test implemented via the tidyr and nortest packages in R, confirming that all conditions met the normality assumption.

The updated analysis maintains the significant differences observed previously, although p-values have shifted slightly. We have now included significance values specifically for the treated conditions with decitabine (while azacytidine showed no significant differences in any comparison, Fig 4A). The difference between the high (H) and low (L) treated conditions remains not significant under this test as was in the previous version of the paper, reflected in Figure 4A and S3.

We believe this rigorous statistical approach strengthens the reliability of our conclusions.

2- I would recommend adding the oxygen consumption data from the other cell lines to Fig4. The authors do include them in a supplemental figure and the text explains that all lines showed the oxygen consumption increase, but I think its important to add them as the assumption made merely by looking at this figure as it stands is that only the high-het line has oxygen consumption increased by methylation inhibitor treatment.

We agree with the reviewer's suggestion and have incorporated the oxygen consumption rate (OCR) data from the other cell lines previously shown in the supplementary figures into the main figure 6. Specifically, Figure 6A now includes OCR data for 13H, WT cybrids, and mouse embryonic fibroblasts (MEFs) harboring intermediate heteroplasmy of the m.5024C>T mutation.

For statistical analysis, we combined the following data from the five Seahorse experiments: three with 13H cybrids, one with WT cybrids, and one with m.5024 MEFs. This approach allowed us to robustly test the effect of decitabine treatment on mitochondrial function across different cell types. Because normalization was performed differently across experiments (using protein content, cell count, or DNA amount), all respiration parameters (basal respiration, ATP production, and maximal respiratory capacity) were normalized relative to their respective control (DMSO) within each experiment (Fig. 6B).

Additionally, we included the proton leak parameter, which showed a slight increase upon treatment. This finding is further supported by the mitochondrial membrane potential measurements added as per Reviewer #3's recommendation (Fig. S5B).

Finally, to complement these findings, we added intracellular ATP measurements assessed via Quinacrine staining in the supplementary section, which confirmed an increase following decitabine treatment (Fig. S5A).

3. Why does methylation inhibition lead to an increase in mitochondrial ROS in all instances? I don't expect the authors to know this, but it's an important observation.

We agree that the increase in mitochondrial ROS following methylation inhibition is an intriguing and important observation. While the precise mechanisms remain to be fully elucidated, we have carefully considered possible explanations in light of existing literature and our data. Previous studies have demonstrated that demethylating agents can impact mitochondrial function^{4,5}, though, to our knowledge, ours is the first study examining these effects in cells harboring mtDNA mutations.

Decitabine has been reported to increase ATP production and oxygen consumption rate (OCR), consistent with our findings^{4,5}. It has also been shown to increase ROS levels and reduce mitochondrial membrane potential⁶—effects we replicated and quantified by measuring mitochondrial membrane potential parameters in this revised version (Fig. S5B), where we observed mild mitochondrial uncoupling with decitabine treatment.

Taken together, our data show increased ATP production, mild uncoupling, elevated OCR, and increased ROS. In the presence of dysfunctional mitochondrial complexes, enhanced electron leakage could lead to increased ROS. Increased ATP synthase (Complex V) activity could underlie these observations—resulting in higher ATP production, mild uncoupling, and elevated OCR.

Although we did not observe increased expression of Complex V subunits (we examined only one of 18 subunits), a previous study in bone marrow nuclear cells showed that ATP5B, a Complex V subunit, is hypermethylated, and that 5-azacytidine increases its mRNA expression⁷. In our EPIC methylation data (Supplementary File 1), only one of the 16 nuclear-encoded Complex V subunits is assessed, and the CpG sites associated with ATP5F show varying methylation levels with no differences across WT, high, and low heteroplasmy cybrids. Nonetheless, it remains possible that decitabine treatment could increase Complex V activity.

Beyond the exact molecular mechanism, these metabolic changes coincide with the loss of pro-survival epigenetic marks that appear critical for the survival of high heteroplasmy cells.

We have revised the paragraph describing mitochondrial function after decitabine treatment and expanded the discussion section to incorporate this possible explanation.

Referee #2 (Remarks for Author):

The corresponding author of this manuscript had previously reported that mitochondrial stress induces epigenetic modifications of nuclear DNA (ref 23). Here, the authors propose that blocking the adaptive nDNA methylation pattern that is triggered by mitochondrial dysfunction (which favours survival of cells with high mtDNA mutation

load), will result in impaired viability of these cells, thus favouring cells with low levels of mtDNA mutation load. Using cultured cybrids with pathogenic mtDNA mutations at different mutation loads the authors investigate the association between the methylation pattern observed in nDNA and the degree of heteroplasmy. After identifying several differentially methylated genes, somehow involved in cell survival, the authors experimentally test how inhibition of DNA methylation (using 5-azacytidine and decitabine) results in reduced survival of cybrids with high mutation load. They finally propose that this strategy could promote a heteroplasmic shift towards lower heteroplasmic load of pathogenic mutations in patients.

While I think that most experiments are convincing and the overall results support their conclusions (which are relevant in the area of mitochondrial genetics), some inaccuracies in the presentation in the results pose some confusion in the interpretation of their observations and in the transparency of the presentation of the results. The main issue relates to the statistical analyses of some experiments. Often (some cases specifically pointed in my comments below) the authors use parametric methods to compare groups with very few replicates. Since the application of parametric methods have specific requirements regarding the distribution of the results analysed (normality and homogeneity of variances, in most cases), its use should be avoided for experiments with so low number of replicates, for which it cannot be determined if data meets these requirements. Nonparametric methods should be used instead. It is true that nonparametric tests are less powerful than parametric tests to detect differences, but the authors can still discuss about their data and the observed values, independently of whether the number of experimental data is sufficient to reach statistical significance or not. In addition, there are several comparisons in which the authors used paired tests for experiments in which the individual results of one group do not seem to be paired with the individual results of the compared group or groups.

We fully agree with the reviewer's insightful comments regarding the limitations imposed by small sample sizes on the use of parametric statistical methods.

To address these concerns thoroughly, we consulted a statistics expert who guided us in re-evaluating our analyses. As a result, we have replaced most parametric tests with appropriate nonparametric alternatives, which do not rely on distributional assumptions and are better suited for datasets with small numbers of replicates.

We also carefully reviewed all instances of paired statistical tests. For each case, we have provided clear explanations justifying the use of paired comparisons where applicable.

We will provide detailed responses addressing each specific example pointed out by the reviewer below.

We appreciate these suggestions, which have helped us improve the rigor and transparency of our statistical analyses.

Also, I could not find anywhere indications about what the error bars state for in the plots. Either the authors should indicate in the methodology section what error bars represent (if it is the same for all plots) or they should explicitly indicate it in each figure legend.

The bars represent the mean \pm SEM. We have added that statement in all the figure legends.

In most cases, the individual results are shown as dots in the plots (with few exceptions, some of them pointed below) and the readers can see the extent of the differences. Forcing the statistics to show significance with use of tests for results that do not meet their requirements is not accurate.

We appreciate the reviewer's emphasis on transparent data presentation. In response, we have added individual data points as dots to all plots, except for Figure 2B, for which we provide a specific explanation below. This visual representation allows readers to directly appreciate the variability and distribution of the data.

Additionally, to meet the journal's requirements for transparency, we have included all p-values directly in the main figures.

Furthermore, following consultation with a statistics expert, we carefully reviewed and addressed each statistical concern raised by the reviewer, as detailed in the point-by-point responses below.

Some specific examples of these points:

-Page 11 (Fig 2): We need to know what the error bars stand for. Using Student t test for $N \geq 2$ is too forced. A nonparametric test should be used. In addition, for so low numbers, bar and error bars is not informative enough. It would be best plotting the actual values as dots (as done in Figs 4, 5, 6 etc).

We agree that clarity in data presentation is essential.

Firstly, the error bars in all figures, including Figure 2, represent the standard error of the mean (SEM). We have now explicitly added this information to all figure legends for clarity.

Regarding Figure 2B, the data represent mean methylation values across 35 CpG sites for each condition. For instance, specifically, for the 13H condition, MS-MLPA analysis was performed on 4 biological replicates, and for each CpG site, the mean methylation value across these replicates was plotted. Thus, the graph is based on 35 mean values per condition. Plotting individual data points for all 35 CpG sites would result in a visually cluttered graph, especially since many values are zero. Therefore, we opted to use bar graphs without individual dots to maintain clarity.

In line with the reviewer's suggestion regarding the statistical analysis, we replaced the previous Student's t-test with a non-parametric Friedman test followed by Dunn's multiple comparison test. To perform this, we excluded 3 CpG sites with missing methylation values in some samples (grey areas in Fig 2A), reducing the dataset to 32 CpGs per condition. This adjustment did not affect the significance of the differences observed.

Another question is: why they use paired analyses? Unless there is an experimental link between each one-to-one specific result paired analyses cannot be used.

In the case of our methylation analysis, we chose paired comparisons because the data for each condition are organized by CpG site. Each CpG site is measured across all experimental groups, making it possible—and statistically appropriate—to assess how methylation at the same CpG site varies between conditions. Since the methylation level at each individual CpG site can differ widely (some are highly methylated, others barely methylated), using a paired approach allows us to control for

this intrinsic variability and better capture consistent changes across conditions. This pairing is not arbitrary but reflects a biologically and experimentally meaningful structure in the dataset: each CpG site serves as its own baseline for comparison across groups.

-Page 15, Fig 4A: Again, it is not clear why the test used to compare different cell lines was a repeated measures ANOVA. The groups that are compared in this figure do not derive from repeated or related measures. ...The only repeated measure I can see here is fluorescence at days 0 and 3, and these two repeated measures are used to calculate the ratio (each data value in the plot). The same applies to Fig S3. Fig S3 should show the individual results, as Fig. 4 does. In addition, I do not think comparing groups of N=3 through a parametric method such as ANOVA is accurate because it cannot be ensured that a sample of 3 results meets the requirements for the use of parametric methods (i.e., derive from a normal distribution and with homogeneous variances among groups). A nonparametric test would be more appropriate. As mentioned above, nonparametric tests are less powerful than parametric tests to detect differences, but the authors can still discuss about their data and the observed values, independently of whether the number of experimental data is sufficient to reach statistical significance or not. For example, for the DMSO group, perhaps nonparametric methods cannot detect significant differences between 13H and 13L (or between 8H and 8L), but the readers will be aware of the data and how the discussion and conclusions are supported or not by the data.

Thank you for this valuable observation. Our initial use of repeated measures ANOVA was based on the fact that all cell lines and treatments were processed in parallel within each independent experiment—cells were plated on the same day and exposed to the same batch of reagents. However, we recognize that this setup may not constitute true repeated measures across groups, and appreciate the clarification.

To address this, we consulted with a statistics expert and revised our analysis accordingly. We now use a two-way ANOVA with fixed categorical variables (cell line and treatment), followed by Tukey's Honest Significant Difference test for post-hoc comparisons. Residual normality was evaluated using the Shapiro-Wilk test via the *tidyr* and *nortest* packages in R. All groups met the normality assumption, and the overall pattern of significant differences remained consistent, with only minor adjustments in p-values.

As also suggested, we revised Figure S3 to include individual data points, ensuring better transparency and interpretability of the dataset. We updated the figure legends to reflect these changes.

-Page 18, Fig 6B. The authors use a parametric test for comparing groups with only N=3. Also related to the 6B plots: why a paired statistical test was used? If there is an experimental link between each DMSO result with a corresponding -dec result (for example, 3 pairs of different one-to-one comparisons done in three different days), then it would be more informative and transparent for readers to link each related pair with a line. If they are independent measurements, then a paired comparison should not be used.

The Seahorse experiments were indeed conducted in matched DMSO–decitabine pairs, using the same cartridge and performed on the same day. This experimental setup justifies the use of paired statistical analyses, as each DMSO–decitabine pair reflects a direct within-experiment comparison.

In response to reviewers' comments, we have now expanded the analysis shown in Figure 6. We included Seahorse data from additional cell conditions that were previously shown in the supplementary material and are now presented in Figure 6A. Quantification in Figure 6B is based on the combined data from all experiments, with results normalized to the DMSO control of each experiment to account for variability across runs and normalization methods. This approach allows a clearer representation of the drug's effect across different cell backgrounds, while maintaining internal consistency. To support the OCR findings, we have also added complementary measurements: intracellular ATP levels assessed using Quinacrine (now included in Fig. S5A) and mitochondrial membrane potential changes upon decitabine treatment (Fig. S5B).

-Page 20, Figure 7A: Again, a paired test is used for an experiment in which each result does not seem to be linked to another specific result of the compared group. When results are linked, the link should be identified for transparency. If the authors consider each individual experiment pairs each 13H-untreated result to a specific 13H-dec result, as one pair of linked results (N=5), why the central column only has N=3 ? The same applies to 7D (6-8 independent experiments).

The comparisons in Figures 7A and 7D were analyzed as paired because each treated sample (13H-aza or 13H-dec) was compared to its corresponding untreated control (13H) within the same experiment and qPCR run. This approach was chosen due to the known temporal variability in mtDNA heteroplasmy levels and potential variability introduced during qPCR. Analyzing these results as paired allows us to control for these sources of variation.

Regarding the unequal number of replicates across treatment groups (e.g., N=5 for 13H and 13H-dec, but N=3 for 13H-aza), this reflects the fact that not all treatments were included in every experiment. For example, in some experiments we compared 13H and 13H-dec only, in others 13H and 13H-aza, and in a few all three conditions were analyzed together. This accounts for the unequal Ns. The same rationale applies to Figure 7D.

We have now clarified this point in the figure legends and ensured that the paired nature of the comparisons is made more transparent to readers. We included a new 13H–13H-aza experiment, which did not alter the statistical significance of the results.

Other comments:

-Page 11 (Fig 2): Methylation levels of 8L and 8H are not different and much lower than WT. How the authors explain this?

Figure 2 focuses on 35 CpG sites within strategic promoter regions of selected tumor suppressor genes (TSGs). In this targeted analysis, '8' cybrids show lower methylation levels than wild-type cybrids, although these differences were not statistically significant. It is likely that these specific TSG regions are not the primary targets of methylation remodeling in the '8' cybrids.

To better capture broader methylation changes, we performed genome-wide analysis using the EPIC 850k array. This revealed that '8' cybrids have overall higher methylation levels than wild-type, though still lower than those observed in '13' cybrids. Moreover, a consistent pattern of higher methylation in high-heteroplasmy (H) versus low-heteroplasmy (L) cybrids was evident across both cybrid types.

Overall, the '13' cybrids exhibit a more pronounced epigenetic remodeling, which appears to play a greater role in shaping their phenotypic behavior. This is reflected throughout the paper in their distinct methylation profiles, more evident differences between H and L conditions, and their stronger dependency on the methylation state for survival.

-The cybrids were always cultured in a medium with high content in glucose. In these conditions, the cybrids entirely rely on glycolysis and pyruvate reduction to lactate, with no functional dependence on the oxidative phosphorylation. Did the authors try to replicate some experiments using galactose instead of glucose? This would make the cells more dependent on OXPHOS, which may have consequences in terms of tolerance and survival of high heteroplasmy of mutations.

This is an insightful observation. We did attempt to culture the cybrids in galactose medium, primarily in the context of selecting for clones with lower heteroplasmy. However, we found that treatment with ethidium bromide was more effective for this purpose. High-heteroplasmy cybrids were unable to survive in galactose-containing medium, and the few cells that did persist did not exhibit lower heteroplasmy levels. In contrast, cybrids with very low heteroplasmy were able to grow under these conditions.

Given that high-heteroplasmy cells are already at a great disadvantage in galactose—where dependence on OXPHOS is increased—we considered that adding demethylating agents would further compromise their viability, making direct comparisons to low-heteroplasmy cells under these conditions challenging.

Minor points:

-Page 2: the authors enumerate: "succinate, fumarate, NAD, NADH and reactive oxygen species". In this context, when they say "NAD", I think they refer to the cationic form NAD⁺, because the acronym "NAD", without the plus sign, means "nicotine amide dinucleotide", which includes both NAD⁺ and NADH. Therefore, I think it should be written "NAD⁺".

Thank you very much for pointing this out, we have changed it to NAD⁺.

-Page 15 (Figure 4A): The authors may consider switching the hierarchy of the X axis label. Since the measured variable of interest is cell proliferation, and the "units" used to quantify cell proliferation is the ratio "fluorescence at day 3/day0" (which are the actual values indicated in the axis numbers), perhaps it would make more sense to label "Proliferation (fluorescence at day 3/day0)". The same applies to Fig. S3.

We have changed the y axis of figures 4A and S3, as suggested.

Referee #3 (Comments on Novelty/Model System for Author):

This manuscript presents a comprehensive and insightful analysis of the epigenetic alterations associated with mitochondrial DNA (mtDNA) mutations. The authors also

provide compelling evidence that interventions aimed at reversing nuclear DNA methylation may lead to a reduction in heteroplasmy levels, both in cellular systems and in vivo models. The experiments investigating DNA methylation are of particularly high quality and contribute meaningfully to the field.

However, some aspects of the study would benefit from further refinement. Specifically, the analyses related to mitochondrial respiration, reactive oxygen species (ROS) production, and respiratory chain protein expression appear to lack appropriate controls, which makes it challenging to fully assess the significance and robustness of these findings.

We appreciate the reviewer's suggestion. To address this, we have included additional controls in all functional assays: for ROS measurements, we used H₂ O₂ as a positive control; for TMRE assays, CCCP was applied to depolarize mitochondria; ATP measurements included CCCP plus oligomycin as a negative control; and UV treatment was used as an apoptosis activator control. These controls have now been incorporated and clearly indicated in all relevant figures.

A major concern pertains to the translational relevance of the proposed therapeutic strategy. As tissues primarily affected in mitochondrial diseases are largely post-mitotic, it remains unclear how inducing apoptosis in high-heteroplasmy cells would confer a therapeutic benefit in such contexts. To strengthen the translational impact of the work, the inclusion of additional experiments using more physiologically relevant or disease-representative models is strongly recommended.

We fully agree with the reviewer's concern regarding the translational relevance of our approach which is addressed in the discussion section. Post-mitotic tissues, which are mainly affected in mitochondrial diseases, present a challenge for strategies based on apoptosis of high-heteroplasmy cells. Nonetheless, our experiments provide evidence that at least in some tissues, epigenetic modulation can lower heteroplasmy.

In the near future, we aim to test this strategy in animal models of mitochondrial disease; however, it is important to acknowledge that such models carrying pathogenic mtDNA mutations at high heteroplasmy are scarce and not widely available. We have addressed this limitation in the discussion section. Combining this approach with strategies that promote regeneration in post-mitotic tissues may be a strategy to achieve therapeutic benefit. This will be a major focus of our future work and hopefully next publication.

Referee #3 (Remarks for Author):

In this study, Perez and colleagues investigate the potential of epigenetic modulators to alter heteroplasmy levels in cybrid models harboring various mtDNA mutations. The authors initially demonstrate that cells with a high proportion of mutant mtDNA acquire a distinctive nuclear DNA methylation profile that is critical for their survival and not observed in cells with low heteroplasmy. Utilizing FDA-approved inhibitors of DNA methylation, the authors selectively compromised the viability of high-heteroplasmy cells, resulting in an enrichment of cells with lower mutant loads and an overall reduction in heteroplasmy. These findings were validated in both cellular and animal models, highlighting a previously underappreciated role of nuclear DNA methylation in supporting the persistence of cells experiencing severe mitochondrial dysfunction. This

work suggests that targeting nuclear epigenetic mechanisms may represent a promising therapeutic avenue for mitochondrial diseases, potentially expanding the current landscape of treatment strategies beyond direct mtDNA manipulation.

While the study is innovative and presents a compelling concept, I have several points of concern, particularly regarding the translational applicability of the proposed intervention. Since mitochondrial diseases primarily affect post-mitotic tissues, it remains unclear how selectively inducing apoptosis in high-heteroplasmy cells would provide therapeutic benefit in such non-proliferative contexts. Additional investigations using more physiologically relevant models could help address this important question.

As mentioned above and discussed in the manuscript's Discussion section, we fully recognize this important limitation. We plan to test our approach in animal models of mitochondrial disease, although such models carrying pathogenic mtDNA mutations are scarce and limited themselves, as it is difficult to achieve high heteroplasmy levels. We are actively seeking appropriate, as this will be a central focus of our future work.

Furthermore, given that the authors' strategy appears to hinge on promoting apoptosis to reduce heteroplasmy, it would be informative to explore whether high-heteroplasmy cells exhibit higher sensitivity to apoptotic stimuli more generally. Experiments utilizing alternative apoptosis inducers could help clarify this point. Along the same lines, it would be of considerable interest to assess the effect of blocking apoptosis-e.g., with Z-VAD-FMK or similar inhibitors-on the efficacy of 5-azacytidine or decitabine in lowering heteroplasmy.

This is a valuable observation and an interesting suggestion, which we have addressed as follows. We exposed 13H and 13L cybrids to an alternative apoptosis inducer, doxorubicin⁸ (5 μ M for 24 hours), and measured apoptosis by Annexin V flow cytometry. Both cybrid types showed increased apoptosis after treatment, although these increases did not reach statistical significance. We relativized the % of apoptotic cells to the control's value (untreated 13H or 13L), (Fig R1). 13H cells tended to be more sensitive to doxorubicin, which is expected as they exhibit lower mitochondrial membrane potential closer to the cytochrome C release threshold.

Figure R1.

Fig R1. Apoptosis assessment of 13H and 13L cells treated with Doxorubicin 5 μ M 24h. % of apoptotic cells were relativized to the untreated respective control. Mann Whitney test was used for comparing 13H-doxo with 13L-doxo. N= 3 experiments for 13H, 4 for 13L.

This could support the concept that pro-apoptotic drugs could strategically target cells with mitochondrial dysfunction. However, we want to emphasize that the increased vulnerability of high-heteroplasmy cells is particularly evident upon downregulation of DNA methyltransferases, especially DNMT1, as shown in Fig 4B. While high-heteroplasmy cells are prone to apoptosis following DNMT1 downregulation, low-heteroplasmy cybrids do not show this susceptibility, highlighting the critical role of the methylation pattern in the survival of high-heteroplasmy cells.

In light of the reviewer's suggestion to distinguish the effects of DNA methylation manipulation from general apoptosis induction, we wanted to specifically highlight the impact of targeting DNA methylation on heteroplasmy. To this end, we mixed 13L cells with 13H cells subjected to inducible DNMT1 knockdown. A 3-day knockdown did not result in a measurable decrease in heteroplasmy (data not shown), so we extended the knockdown to one week. In three experiments, two showed a decrease in heteroplasmy (Fig R2), supporting the main hypothesis. In the third experiment, where no reduction was observed, DNMT1 downregulation was not sustained (Fig R3), which could explain the lack of effect.

These findings indicate that inducible DNMT1 knockdown cannot be reliably maintained long enough to reduce heteroplasmy effectively. Therefore, we believe DNA methylation modulators—especially decitabine—offer greater translational promise, as it: 1) reduces methylation more efficiently; 2) induces apoptosis more strongly in high-heteroplasmy cells; 3) decreases heteroplasmy more consistently; 4) is suitable for animal model testing; and 5) is FDA-approved for potential clinical use. Our view is that decitabine's effect on high-heteroplasmy cells likely involves both its demethylating and metabolic actions

Figure R2.

Fig R2. m.13513G>A heteroplasmy variation when DNMT1 is down-regulated in 13H cybrids. A mixture of 13L+13H(modified with the inducible shDNMT1 construct) were plated. DNMT1 was down-regulated using IPTG. After 7 days, heteroplasmy was measured showing a reduction in 2 out of 3 experiments. Paired comparisons are shown with dotted lines.

Figure R3.

Fig R3. DNMT1 expression of 13H cells after 7 days with IPTG treatment. DNMT1 expression returned to the basal control state, feature that can explain the inability to reduce heteroplasmy consistently.

Regarding the suggestion to block apoptosis induced by decitabine and assess the impact on heteroplasmy, we conducted experiments using two apoptosis inhibitors: Z-VAD-FMK and calpeptin⁹. The inhibitors were added 2 hours prior to decitabine treatment. Specifically, Z-VAD-FMK (100 μ M) was combined with decitabine (1 μ M) for 3 days on 13H cells, and calpeptin (10 μ M) was applied on 13H+13L mixtures alongside decitabine. We confirmed that both inhibitors effectively blocked decitabine-induced apoptosis (Fig R4, left panel). Subsequently, we evaluated heteroplasmy levels (Fig R4, right panel) and observed that in the presence of apoptosis inhibitors, decitabine failed to reduce heteroplasmy. Although significance was not reached due to limited sample sizes (N=4 and N=3), the trend was consistent. In one experiment with 13H+13L, decitabine did not reduce heteroplasmy and calpeptin did not modify this. These findings support that the reduction of heteroplasmy by decitabine is mediated primarily through selective apoptosis of high-heteroplasmy cells. **Figure R4.**

Fig R4. Apoptosis (left panel) and heteroplasmy (right panel) determination in response to decitabine and decitabine combined with apoptosis inhibitors z-vad-fmk in 13H cells (upper panels) and calpeptin in mixture of 13H and 13L cells (lower panels). Wilcoxon matched-pairs signed rank test. Paired comparisons are shown with dotted lines.

I also have some concerns regarding the analysis of mitochondrial function. In Figures 6 and S4, the authors report that decitabine increases oxygen consumption in wild-type (WT) cybrids, MEFs, and 13H cells. It would enhance interpretability if data from WT and 13H cybrids were plotted on the same graph to facilitate direct comparison. Moreover, clarification is needed on what is meant by "WT cybrids"—does this refer to WT 143B cells, or to cybrids with 0% heteroplasmy?

This is an excellent suggestion, as plotting all three groups together facilitates a clearer interpretation of decitabine's overall metabolic effects. Accordingly, we have now included 13H, WT, and m.5024 MEFs in the main Fig 6. These combined data show that decitabine similarly increases oxygen consumption and related parameters—basal respiration, ATP production, and maximal respiratory capacity—across all cell lines and heteroplasmy levels. Additionally, we included the proton leakage parameter, which showed a slight increase, and this was further supported by a reduction in mitochondrial membrane potential measurements added as suggested by yourself (Fig S5B).

Furthermore, we added intracellular ATP measurements using Quinacrine in the supplementary material (Fig S5A), which are consistent with the OCR data and confirm increased ATP levels upon decitabine treatment.

Regarding the definition of "WT cybrids," these refer to cybrids with 0% mutant mtDNA, not to WT 143B cells.

The reported increase in respiration and ATP production appears somewhat inconsistent with the concurrent rise in ROS, as improved mitochondrial coupling would typically be associated with decreased ROS generation. It would be valuable to know whether the authors measured mitochondrial membrane potential to support their interpretations.

We understand that the observed effects on the respiratory chain may seem paradoxical, as some parameters indicate improved mitochondrial function while others suggest the opposite. To further support our interpretations, we measured mitochondrial membrane potential and found a slight decrease following decitabine treatment, which we have now included in the supplementary material (Fig S5B).

Previous studies have reported that demethylating agents can affect mitochondrial function. To our knowledge, however, this is the first time such agents have been tested in cells harboring mtDNA mutations.

Literature shows that decitabine can increase ATP production and oxygen consumption rate (OCR)^{4,5}, while also increasing ROS production and reducing mitochondrial membrane potential⁶, consistent with our observations.

Considering the potential metabolic impact of demethylating agents in our model, we would like to emphasize the following: we see an increased ATP production, mild decoupling and increased OCR and ROS. In our system, electron leakage from dysfunctional mitochondrial complexes may contribute to the increased ROS. Increased activity of ATP synthase (Complex V) could potentially contribute to this pattern, resulting in higher ATP production alongside mild uncoupling and increased OCR. Although we did not observe increased expression of Complex V (we assessed only one of 18 subunits), previous work in bone marrow cells showed that one Complex V subunit (ATP5B) is hypermethylated and that 5-azacytidine treatment upregulates its

mRNA expression⁷. There is limited evidence of the methylation status of these genes in 143B cells. In our EPIC methylation data (Supplementary File 1), only one of the 16 nuclear-encoded Complex V subunits was covered. Among six CpG sites related to ATP5F, methylation levels varied (four low, one intermediate, one high), with no differences between wild-type, high, or low heteroplasmy cybrids. Nonetheless, decitabine could be influencing Complex V activity.

Beyond the precise mechanism of this metabolic derangement with decitabine treatment in our model, it occurs alongside the loss of pro-survival epigenetic marks important for the high heteroplasmy cells.

We have updated the relevant paragraph on mitochondrial function and added to the discussion section addressing this potential explanation.

Lastly, the analysis of respiratory chain proteins by western blot could be further strengthened by including the corresponding wild-type (or low-heteroplasmy) controls for MEFs and 13H cells, thereby providing a more complete and comparative perspective.

We have measured mitochondrial respiratory chain complex proteins in both 13L and 13L-dec cells and did not observe any differences upon decitabine treatment. These data have now been included in the supplementary material (Fig S5C).

Minor comments:

In the Introduction, the statement "We hypothesized that manipulating the nDNA methylome would negatively impact the high-heteroplasmy cells, favor the prevalence of low heteroplasmy ones and consequently, reduce heteroplasmy" is somewhat unclear. A rephrasing for clarity is recommended.

We agree with the reviewer and have rephrased it as: "We hypothesized that altering the nDNA methylome would impair high-heteroplasmy cells, promote the dominance of low-heteroplasmy cells, and thereby lead to an overall reduction in heteroplasmy"

The abbreviations 13H/L and 8H/L used throughout the manuscript are a bit confusing and could be more clearly defined or replaced with more intuitive nomenclature to aid the reader.

We understand that the abbreviations 13H/L and 8H/L may be confusing to some readers. However, after careful consideration, we found no alternative nomenclature that better captures the key differences between these cybrid lines while keeping the terms concise. To aid clarity, we have ensured that these abbreviations are clearly defined at their first appearance in the manuscript.

REFERENCES

1. Li, W. *et al.* 5-Azacytidine suppresses EC9706 cell proliferation and metastasis by upregulating the expression of SOX17 and CDH1. *Int J Mol Med* **38**, 1047–1054 (2016).

2. Estekizadeh, A. *et al.* 5-Azacytidine treatment results in nuclear exclusion of DNA methyltransferase-1, as well as reduced proliferation and invasion in human cytomegalovirus-infected glioblastoma cells. *Oncol Rep* **41**, 2927–2936 (2019).
3. Ong, A. L. C., Lee, S. H., Aung, S. W., Khaing, S. L. & Ramasamy, T. S. 5-Azacytidine pretreatment confers transient upregulation of proliferation and stemness in human mesenchymal stem cells. *Cells and Development* **165**, (2021).
4. Ni, X. *et al.* Low-dose decitabine modulates myeloid-derived suppressor cell fitness via LKB1 in immune thrombocytopenia. *Blood* **140**, 2818–2834 (2022).
5. Zhu, X. ying *et al.* AICAR and Decitabine Enhance the Sensitivity of K562 Cells to Imatinib by Promoting Mitochondrial Activity. *Curr Med Sci* **40**, 871–878 (2020).
6. Fandy, T. E. *et al.* Decitabine induces delayed reactive oxygen species (ROS) accumulation in leukemia cells and induces the expression of ROS generating enzymes. *Clinical Cancer Research* **20**, 1249–1258 (2014).
7. Yang, J. *et al.* Hypermethylation of CpG sites at the promoter region is associated with deregulation of mitochondrial ATPsyn-? and chemoresistance in acute myeloid leukemia. *Cancer Biomarkers* **16**, 81–88 (2016).
8. Huigsloot, M., Tijdens, I. B., Mulder, G. J. & Van Water, B. De. Differential regulation of doxorubicin-induced mitochondrial dysfunction and apoptosis by Bcl-2 in mammary adenocarcinoma (MTLn3) cells. *Journal of Biological Chemistry* **277**, 35869–35879 (2002).
9. Wood, D. E. & Newcomb, E. W. Caspase-dependent activation of calpain during drug-induced apoptosis. *Journal of Biological Chemistry* **274**, 8309–8315 (1999).

2nd Jul 2025

Dear Dr. Mayorga,

Thank you for the submission of your revised manuscript to EMBO Molecular Medicine. I am pleased to inform you that we will be able to accept your manuscript pending the following final amendments:

1) Please address the referee #2 and #3 concerns by clearly acknowledging the limitation of the study.
2) Figures: Please remove all main figures from the manuscript file and upload them as individual high-resolution files in TIFF, EPS or PDF format. Supplementary figures should also be removed from the manuscript file, renamed to Figure EV1 etc. and uploaded as individual high-resolution files in TIFF, EPS or PDF format. Please update their callouts in the main manuscript text. Main figure legends should be moved to the end of the manuscript followed by EV Figure legends. Please check "Author Guidelines" for more information:

<https://www.embopress.org/page/journal/17574684/authorguide#figureformat>

<https://www.embopress.org/page/journal/17574684/authorguide#expandedview>

3) Please address all comments suggested by our data editors listed below:

o Figure legends:

1. Please note that the exact p values are not provided in the legends of figures 3B, S1, S3, S4A, B; S6A, B.

2. Please note that information related to n is missing in the legends of figures 3B, S5 B.

- Correct order of manuscript sections: Abstract / Keywords / The Paper Explained / Introduction / Results / Discussion / Methods / Data Availability / Acknowledgements / Disclosure and Competing Interests Statement / References / Main Figure Legends / Tables / Expanded View Figure Legends.

- Add up to 5 keywords.

- Figure callouts should be in a sequential order. Currently, Fig 8 is called out before Figure 2B. Please correct.

- Rename "Declaration of competing interest" to "Disclosure and competing interests statement". We updated our journal's competing interests policy in January 2022 and request authors to consider both actual and perceived competing interests. Please review the policy <https://www.embopress.org/competing-interests> and update your competing interests if necessary.

- Author contributions: Please remove it from the manuscript and specify author contributions in our submission system. CRediT has replaced the traditional author contributions section because it offers a systematic machine-readable author contributions format that allows for more effective research assessment. You are encouraged to use the free text boxes beneath each contributing author's name to add specific details on the author's contribution. More information is available in our guide to authors:

<https://www.embopress.org/page/journal/17574684/authorguide#authorshippinguidelines>

- In Methods, provide the statement that informed consent was obtained from all human subjects and confirm that the experiments conformed to the principles set out in the WMA Declaration of Helsinki and the Department of Health and Human Services Belmont Report.

- In Methods, add the following paragraph:

Graphics:

(some of the... OR Figure #... OR synopsis) Graphics were created with BioRender.com/Corel draw/etc.

- Indicate in legends exact n and exact p values, not a range, along with the statistical test used. To keep the figures "clear" some authors found providing an Appendix table Sx with all exact p-values preferable. You are welcome to do this if you want to.

- In Methods, a statistical paragraph should reflect all information that you have filled in the Authors Checklist, especially regarding randomization, blinding, replication.

- Please include structured Methods section that includes a Reagents and Tools Table (should be uploaded as a separate file) followed by a Methods and Protocols section. More information on how to adhere to this format as well as downloadable templates (.docx) for the Reagents and Tools Table can be found in our author guidelines:

<https://www.embopress.org/page/journal/17574684/authorguide#structuredmethods>

An example of a paper with Structured Methods can be found here:

<https://www.embopress.org/doi/full/10.1038/s44320-024-00037-6#sec-4>

- Please use the following format to report the accession number of your data:

[data type]: [full name of the resource] [accession number/identifier] ([doi or URL or identifiers.org/DATABASE:ACCESSION])

Please check "Author Guidelines" for more information.

<https://www.embopress.org/page/journal/17574684/authorguide#availabilityofpublishedmaterial>

- Please correct the reference citation in the text and reference list. In the text a reference should be cited by author and year of publication. Include a space between a word and the opening parenthesis of the reference that follows. In the reference list, citations should be listed in alphabetical order. Where there are more than 10 authors on a paper, 10 will be listed, followed by "et al.". Also, please remove DOIs. DOIs should only be used for preprints and datasets that have not been published. Please check "Author Guidelines" for more information.

<https://www.embopress.org/page/journal/17574684/authorguide#referencesformat>

4) Supplementary files: Rename supplementary files to Dataset EV1 etc. and update their callouts in the main manuscript text. Please note that supplementary file 3 is missing.

5) Funding: Please merge it with "Acknowledgements". Also, make sure that information about all sources of funding are complete in both our submission system and in the manuscript. Wood-Whelan research fellowship 2019 for a short stay at Moraes Lab (Miller School of Medicine, University of Miami, USA is missing in our submission system.

6) The Paper Explained: Please provide "The Paper Explained" and add it to the main manuscript text. Please check "Author Guidelines" for more information. <https://www.embopress.org/page/journal/17574684/authorguide#researcharticleguide>

7) Synopsis: Every published paper now includes a 'Synopsis' to further enhance discoverability. Synopses are displayed on the journal webpage and are freely accessible to all readers. They include separate synopsis image and synopsis text.

- Synopsis image: Please provide a visual abstract as a high-resolution jpeg file 550 px-wide x 300-600 pixels high to illustrate your article.

- Synopsis text: Please provide a short standfirst (maximum of 300 characters, including space) as well as 2-5 one sentence bullet points that summarise the paper as a .doc file. Please write the bullet points to summarise the key NEW findings. They should be designed to be complementary to the abstract - i.e. not repeat the same text. We encourage inclusion of key acronyms and quantitative information (maximum of 30 words / bullet point). Please use the passive voice.

8) As part of the EMBO Publications transparent editorial process initiative (see our Editorial at <http://embomolmed.embopress.org/content/2/9/329>), EMBO Molecular Medicine will publish online a Review Process File (RPF) to accompany accepted manuscripts. This file will be published in conjunction with your paper and will include the anonymous referee reports, your point-by-point response and all pertinent correspondence relating to the manuscript. Let us know whether you agree with the publication of the RPF and as here, if you want to remove or not any figures from it prior to publication. Please note that the Authors checklist will be published at the end of the RPF.

9) Please provide a point-by-point letter INCLUDING my comments as well as the reviewer's reports and your detailed responses (as Word file).

I look forward to reading a new revised version of your manuscript as soon as possible.

Yours sincerely,

Zeljko Durdevic

Zeljko Durdevic
Senior Editor
EMBO Molecular Medicine

*** Instructions to submit your revised manuscript ***

1) a .docx formatted version of the manuscript text (including Figure legends and tables)

2) Separate figure files*

3) supplemental information as Expanded View and/or Appendix. Please carefully check the authors guidelines for formatting

Expanded view and Appendix figures and tables at
<https://www.embopress.org/page/journal/17574684/authorguide#expandedview>

4) a letter INCLUDING the reviewer's reports and your detailed responses to their comments (as Word file).

5) The paper explained: EMBO Molecular Medicine articles are accompanied by a summary of the articles to emphasize the major findings in the paper and their medical implications for the non-specialist reader. Please provide a draft summary of your article highlighting

6) Author contributions: the contribution of every author must be detailed in a separate section.

7) EMBO Molecular Medicine now requires a complete author checklist (<https://www.embopress.org/page/journal/17574684/authorguide>) to be submitted with all revised manuscripts. Please use the checklist as guideline for the sort of information we need WITHIN the manuscript. The checklist should only be filled with page numbers where the information can be found. This is particularly important for animal reporting, antibody dilutions (missing) and exact values and n that should be indicated instead of a range.

8) Every published paper now includes a 'Synopsis' to further enhance discoverability. Synopses are displayed on the journal webpage and are freely accessible to all readers. They include a short stand first (maximum of 300 characters, including space) as well as 2-5 one sentence bullet points that summarise the paper. Please write the bullet points to summarise the key NEW findings. They should be designed to be complementary to the abstract - i.e. not repeat the same text. We encourage inclusion of key acronyms and quantitative information (maximum of 30 words / bullet point). Please use the passive voice. Please attach these in a separate file or send them by email, we will incorporate them accordingly.

You are also welcome to suggest a striking image or visual abstract to illustrate your article. If you do please provide a jpeg file 550 px-wide x 300-600px high.

9) A Conflict of Interest statement should be provided in the main text

10) Please note that we now mandate that all corresponding authors list an ORCID digital identifier. This takes <90 seconds to complete. We encourage all authors to supply an ORCID identifier, which will be linked to their name for unambiguous name identification.

Currently, our records indicate that the ORCID for your account is 0000-0001-7453-4919.

Please click the link below to modify this ORCID:
Link Not Available

11) Include a Reagents and Tools Table as part of the Methods section, which can be downloaded from our author guidelines (<https://www.embopress.org/page/journal/17574684/authorguide#structuredmethods>)

- Graphs 800-1,200 DPI
- Photos 400-800 DPI
- Colour (only CMYK) 300-400 DPI"

*Additional important information regarding figures and illustrations can be found at
<https://bit.ly/EMBOPressFigurePreparationGuideline>. See also figure legend preparation guidelines:
<https://www.embopress.org/page/journal/17574684/authorguide#figureformat>

***** Reviewer's comments *****

Referee #2 (Comments on Novelty/Model System for Author):

The transparency of the statistical analyses has improved as compared with the first version of the manuscript. Although I still have some concerns about how robust are results attributing normal distribution to samples of $N=3$, now the readers have all the information in the report, so they have the elements to critically read the report. The manuscript reports some novel experimental studies on observations previously reported by the authors, with a moderate medical impact given that the proliferative studied models are different from quiescent tissues usually affected in mitochondrial disorders. However, the models are adequate in the context of the planned experimental aims and for the specific variables they study, and the authors explicitly discuss this limitation in the manuscript. Overall, the results are interesting and informative, and I think the report deserves publication.

Referee #2 (Remarks for Author):

No additional remarks to the authors

Referee #3 (Comments on Novelty/Model System for Author):

Although I acknowledge the effort the authors put in clarifying several issues and strengthening their hypothesis, I am still not convinced that the cell lines and models they used are the best to support their conclusions.

Referee #3 (Remarks for Author):

The authors clarified my main comments. Although I still have concerns about the real impact of these findings for mitochondrial diseases, I have no further comments at this stage.

The authors addressed the remaining editorial issues.

17th Jul 2025

Dear Dr. Mayorga,

We are pleased to inform you that your manuscript is accepted for publication and is now being sent to our publisher to be included in the next available issue of EMBO Molecular Medicine.

Zeljko Durdevic
Senior Editor
EMBO Molecular Medicine
